METHODS AND RESOURCES

# A single-cell atlas of the sexually dimorphic *Drosophila* foreleg and its sensory organs during development

**Ben R. Hopkins** [ID]*, **Olga Barmina, Artyom Kopp**

Department of Evolution and Ecology, University of California, Davis, California, United States of America

* brhopkins@ucdavis.edu

**Data Availability Statement:** Raw sequencing reads and preprocessed sequence data are available through NCBI GEO with accession code GSE215073. All code used in this paper is available on the Open Science Framework with DOI 10.

## Abstract

To respond to the world around them, animals rely on the input of a network of sensory organs distributed throughout the body. Distinct classes of sensory organs are specialized for the detection of specific stimuli such as strain, pressure, or taste. The features that underlie this specialization relate both to the neurons that innervate sensory organs and the accessory cells they comprise. To understand the genetic basis of this diversity of cell types, both within and between sensory organs, we performed single-cell RNA sequencing on the first tarsal segment of the male *Drosophila melanogaster* foreleg during pupal development. This tissue displays a wide variety of functionally and structurally distinct sensory organs, including campaniform sensilla, mechanosensory bristles, and chemosensory taste bristles, as well as the sex comb, a recently evolved male-specific structure. In this study, we characterize the cellular landscape in which the sensory organs reside, identify a novel cell type that contributes to the construction of the neural lamella, and resolve the transcriptomic differences among support cells within and between sensory organs. We identify the genes that distinguish between mechanosensory and chemosensory neurons, resolve a combinatorial transcription factor code that defines 4 distinct classes of gustatory neurons and several types of mechanosensory neurons, and match the expression of sensory receptor genes to specific neuron classes. Collectively, our work identifies core genetic features of a variety of sensory organs and provides a rich, annotated resource for studying their development and function.

## Introduction

All behavior rests upon the ability of animals to detect variation in the internal and external environments. In multicellular animals, the detection of such variation is a function performed by sensory organs. With much of its external surface covered by many different classes of sensory organs, the fruit fly *Drosophila melanogaster* has long been used to investigate the mechanisms through which animals sense the world around them. Much attention has focused on the eyes, antennae, and maxillary palps, but the male *Drosophila* forelegs, which perform wide-ranging roles in locomotion, grooming, and courtship (e.g., [1–3]), display a distinct repertoire

17605/OSF.IO/BA8TF (https://www.osf.io/ba8tf/). Processed Seurat objects for each dataset (including all subsetted, annotated datasets) used in this study are available at the same address. A complete protocol for generating single-cell suspensions from pupal tarsi is available through protocols.io at dx.doi.org/10.17504/protocols.io.x54v9dzbmg3e/v1.

**Funding:** This work was supported by the Human Frontier Science Program Organization (LT000123/2020-L) to BRH and National Institutes of Health (R35 GM122592) to AK, and shared instrumentation grants S10OD026702 to the UC Davis Light Microscopy Imaging Facility and 1S10OD010786-01 to the DNA Technologies and Expression Analysis Core at the UC Davis Genome Center. The funders had no role in study design, data collection and analysis, decision to publish, or preparation of the manuscript.

**Competing interests:** The authors have declared that no competing interests exist.

**Abbreviations:** APF, after puparium formation; DEG, differentially expressed gene; FCA, Fly Cell Atlas; FGF, fibroblast growth factor; GRN, gustatory receptor neuron; IR, ionotropic receptor; MSNCB, mechanosensory neuron in chemosensory bristle; PC, principal component; RT-PCR, reverse transcription PCR; scRNA-seq, single-cell RNA sequencing; SOP, sensory organ precursor.

of sensory organs (Fig 1A and 1B). Internal mechanosensory receptors known as chordotonal organs sense proprioceptive stimuli around leg joints [4] and substrate-borne vibrations [5] (Fig 1A). Campaniform sensilla, singly innervated, shaftless sensors embedded in the cuticle, detect and relay cuticular strain, allowing for posture and intraleg coordination to be maintained [6–8] (Fig 1C). Mechanosensory bristles line the surface of the leg, detecting contact by deflection of an external, hair-like process [9,10] (Fig 1D). This organ class isn't uniform: In the males of a subset of *Drosophila* species, including *D. melanogaster*, some mechanosensory bristles are heavily modified to generate a "sex comb," an innovation critical for male mating success [11–14] (Fig 1B). Finally, the foreleg also contains chemosensory taste bristles, which are sexually dimorphic in number and innervated by both multiple gustatory receptor neurons (GRNs) and a single mechanosensory neuron [15–17] (Fig 1E). As in other parts of the body, such as the labellum (e.g., [18]), a degree of functional diversity exists between chemosensory taste bristles on the legs. The tunings and sensitivities of these bristles to a wide panel of tastants vary in relation to both the pair of legs on which they're housed and their position within a given leg [19]. This variation is, at least in part, achieved by restricting the expression of certain gustatory receptors to subsets of taste bristles [3,19]. A level below the bristles themselves, the multiple GRNs that innervate each bristle appear to perform distinct functions. Three distinct GRN classes involved in the detection and evaluation of conspecifics have been resolved in the leg, each of which makes a critical contribution to normal sexual behavior [20–22]. Ultimately, unraveling how this varied sensory apparatus is constructed through development and identifying the molecular basis of specialization in each sensory organ remain central objectives of developmental neurobiology.

The function of sensory organs and their tuning to particular stimuli is not only a product of the neurons that innervate them. In each case, the sensory organ is a composite of multiple distinct cell types and dependent upon the involvement of glia to effectively relay detected signals to the brain. Different organ classes appear to share a common developmental blueprint, such that, despite variation in their form and function, mechanosensory bristles, chemosensory taste bristles, and campaniform sensilla each contain 4 homologous cell types [23]. In the bristle lineage, these are the neurons, which may vary in number between different sensory organ classes (such as between the polyinnervated chemosensory and monoinnervated mechanosensory bristles), along with 3 sensory support cells: the trichogen (shaft or, in campaniform sensilla, the dome), tormogen (socket), and thecogen (sheath). These sensory support cells bear features that clearly define the sensory capabilities of the organ. For example, the elongated shafts of mechanosensory bristles support the deflection-based mechanism through which stimuli are detected [10], the pore at the tip of chemosensory taste bristles enables the receipt of nonvolatile compounds [18], and the elliptical shape of many campaniform sensilla confers sensitivity to the direction of cuticular compression and strain ([24]; reviewed by [25]). But beyond these morphological features, our understanding of the wider, organ-specific contributions that support cells make to the specific sensory capabilities of each organ class remains poor [26]. Yet there is clear potential for their broader involvement in defining an organ type's capabilities, given both their close physical associations with the neurons and, at least in taste bristles, their role in producing the lymph fluid that bathes the dendrites of GRNs and which is central to tastant detection [27,28].

High-throughput single-cell RNA sequencing (scRNA-seq) technologies allow for the transcriptional profiles of many thousands of cells to be recorded from a single tissue. The advent of these technologies has precipitated an explosion of interest in cell type–specific patterns of gene expression. Over the last few years, "atlases" describing the cellular diversity of tissues [29–32], embryos (e.g., [33–35]), and whole adult animals (e.g., [36–38]) have been published for a variety of species. Through such work, regulators of development have been identified

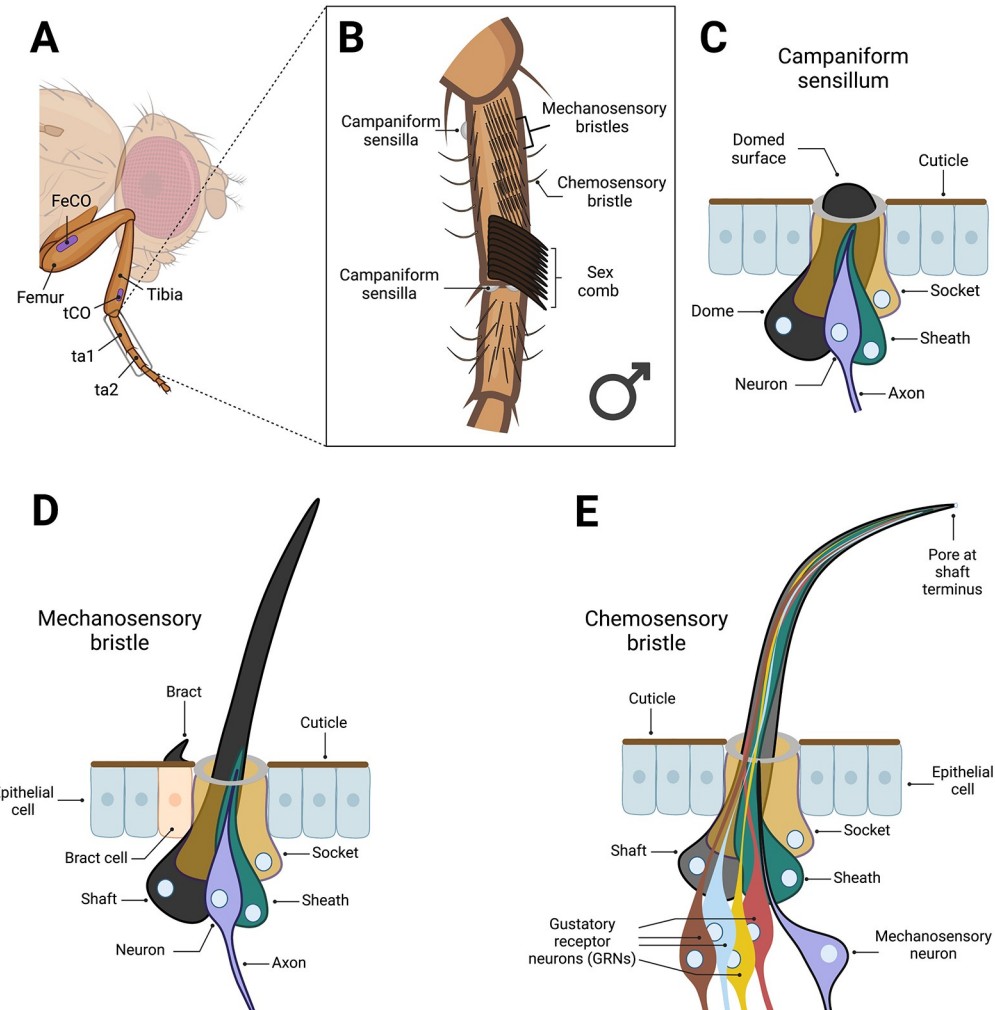

**Fig 1. The first tarsal segment of the male *Drosophila melanogaster* foreleg carries multiple functionally and structurally distinct sensory organs.** (A) Anatomy of the *Drosophila melanogaster* foreleg. The first tarsal segment (ta1), the focal region of this study, is distal to the tibia. Two chordotonal organs (COs) are present outside of the tarsal segments (approximate positions shown in purple). One is situated in the proximal femur (FeCO) and the other in the distal tibia (tCO) [4,5]. (B) The ta1 of the *D. melanogaster* foreleg is enriched for a range of functionally and structurally diverse sensory organs. This region has the highest concentration of mechanosensory bristles of any part of the leg. Here, mechanosensory bristles are arranged in transverse rows on the ventral side, an arrangement thought to aid in grooming, and longitudinal rows on the anterior, dorsal, and posterior sides [9]. In males, the most distal transverse bristle row is transformed into the sex comb: The mechanosensory bristles, now "teeth," are modified to be thicker, longer, blunter, and more heavily melanized, while the whole row is rotated 90° [11,12]. Males also show a sex-specific increase in the number of chemosensory taste bristles in ta1, bearing approximately 11 compared to the female's approximately 7 [16]. Three campaniform sensilla are present in ta1, two on the dorsal distal end of ta1 and one on the proximal ventral side [Ta1GF and Ta1SF, respectively, using the nomenclature of [8]; no campaniform sensilla are present in the distal tibia, ta2, or proximal ta3. (C-E) Campaniform sensilla, mechanosensory bristles, and chemosensory bristles are all composed of modified versions of four core cell types: a socket (or "tormogen"), shaft/dome (or "trichogen"), sheath (or "thecogen"), and neuron [186]. The shaft and socket construct the external apparatus that provides the point of contact for mechanical or chemical stimuli and form a subcuticular lymph cavity that provides the ion source for the receptor current [142,187]. The sheath has glia-like properties, ensheathing the neuron and, as is thought, providing it with protection [187]. Ultimately, however, the contributions of these nonneuronal cells to sensory processing remain poorly characterized [26]. (C) Campaniform sensilla detect strain in the cuticle. They are singly innervated and capped with a dome, rather than a hair-like projection, which extends across the surface of the socket cell [25]. The dendrite tip attaches to the dome cuticle [187]. (D) Mechanosensory bristles detect deflection of the hair-like projection. They are innervated by a single neuron, the dendritic projections of which terminate at the base of the shaft. Specific to this bristle class, the most proximal epithelial cell to the developing sense organ is induced to become a bract cell [12,80]. Bract cells secrete a thick, pigmented, hair-like, cuticular protrusion. (E) The chemosensory taste bristles of the leg differ in their morphology from mechanosensory bristles,

appearing less heavily melanized and more curved. They also house a pore at the terminus of the shaft and lack bracts. Each is innervated by a single mechanosensory neuron and 4 gustatory receptor neurons (GRNs) [16]. Figure created using Biorender.com.

(e.g., [39,40]), novel cell types described (e.g., [41,42]), and the effects of age (e.g., [43,44]) and infection (e.g., [45,46]) on the gene expression profiles of individual cell types characterized. But in arthropods, tissues associated with the cuticle have presented a challenge to single-cell approaches because the cuticle prevents isolation of single cells from peripheral tissues without significant damage [38,47]. Although single-nuclei RNA-seq methods have been used in such tissues (e.g., [38]), scRNA-seq approaches generally offer substantially greater read and gene detection with reduced "gene dropout" and lower expression variability between cells [48]. Consequently, approaches that help characterize the transcriptomes of cuticle-associated cells are of significant value. In principle, the developmental window between pupa–adult apolysis —the separation of the pupal cuticle from the epidermis—and the formation of the adult cuticle (approximately 12 to 48 h after puparium formation (APF)) [49] provides a rare opportunity during which such cells might be accessible.

Here, we use scRNA-seq to profile the sensory organs of the male *D. melanogaster* foreleg at 2 developmental time points that follow soon after the specification of sensory organ cells (24 h and 30 h APF) [11]. Using a fine-scale dissection technique, we specifically target the first tarsal segment (the "basitarsus") to maximize the detection of rare sensory organ types, including the campaniform sensilla, chemosensory taste bristles, and sex comb teeth. We begin by examining the transcriptomic landscape of the tissues in which the sensory organs reside, constructing a spatial reference map of epithelial cells based on intersecting axes of positional marker expression, resolving joint-specific gene expression, and characterizing the distinct repertoires of expressed genes in tendon cells, hemocytes, and bract cells. We then focus on the nonneuronal component of the nervous system, describing the complement of glial cells present in the region, identifying and visualizing wrapping glia, surface glia, and a novel axon-associated cell population that is negative for the canonical glia marker *repo* and appears to contribute toward the construction of the neural lamella. We then resolve and validate a combinatorial transcription factor code unique to the neurons of each of mechanosensory bristles, campaniform sensilla, chordotonal organs, and the sex comb. We further identify and validate a transcription factor code unique to 4 transcriptomically distinct GRN classes, including known male- and female-pheromone sensing neurons, and recover this same code in a published adult leg dataset. With these annotations in place, we link a wide range of genes, including receptors and membrane channels, to specific neuron classes. Finally, we detail the transcriptomic differences that distinguish between sensory organ support cells, both within a single organ class (e.g., sheaths versus sockets) and between classes (e.g., chemosensory sheaths versus mechanosensory sheaths).

## Results

### Homologous clustering of 24 h and 30 h transcriptomes

We generated 2 scRNA-seq datasets from freshly dissected male first tarsal segments using 10× Chromium chemistry. One sample comprised males collected at 24 h APF and the other at 30 h APF. After filtering based on cell-level quality control metrics (see Materials and methods), we recovered 9,877 and 10,332 cells in our 24 h and 30 h datasets, respectively (S1A Fig). In the 24 h dataset, the median number of genes and transcripts detected per cell was 2,083 and 11,292, respectively (S1B and S1D Fig). The equivalent values for the 30 h dataset were 1,245

and 5,050 (S1C and S1E Fig). We began our analysis by constructing separate UMAP plots for each dataset. The clustering pattern of the 30 h dataset largely recapitulated that of the 24 h dataset, which is to say that each cluster in the 24 h dataset had a clear homolog in the 30 h dataset (and vice versa) based on marker gene expression (Fig 2A–2J). Given this concordance, we opted to integrate our 2 datasets and subcluster our data to facilitate closer analysis of rare cell subtypes (see Materials and methods). First, we subclustered epithelial cells and then separated them into joint and nonjoint datasets (Fig 2K). Epithelial cells represented the major cell type in both datasets (70.7% in 24 h: joints = 16.5%, nonjoints = 54.1%; 70.2% in 30 h: joints = 13.6%, nonjoints = 56.5%; Fig 2L). We then subclustered the nonepithelial cells (Fig 2M) and separated them into neurons (based on the expression of *fne*; Fig 2O), sensory support cells (based on the expression of *pros*, *nompA*, *Su(H)*, and *sv*; Fig 2Q), and the remaining nonsensory cells (Fig 2P). The relative proportions of cells in each of these classes was similar between the 2 datasets (24 h: 19.5% neurons, 38.8% sensory support, 41.8% nonsensory; 30 h: 17.1% neurons, 37.0% sensory support, 45.9% nonsensory; Fig 2N). We include our final annotations in Fig 2 and work through the supporting evidence throughout this paper, with an emphasis on sensory and glial cell types.

## Transcriptomic divergence between the tibia/tarsus and intertarsal joints

Joint cells separated from the main body of epithelial cells in our initial epithelial clustering analysis (circled in Fig 3A and 3B). These cells showed enriched expression of genes with known involvement in the formation of or localization at joints, including *drm*, *nub*, and *TfAP-2* (Fig 3A and 3B) [50–52]. After subclustering these cells (Fig 3C and 3D), we used the nonoverlapping expression of *nub* and *TfAP-2* to identify subregions of the joint. We observed expression of GFP-tagged TfAP-2 [53] in both the tibia/tarsal and intertarsal joints, with somewhat weaker staining in the former (Fig 3E). In contrast, expression of *nub-GAL4* was restricted to the joint-adjacent region of the distal tibia (Fig 3E) (see also [54,55]). One of our joint clusters was *TfAP-2*+/*nub*− and additionally positive for *bab2*, a gene with a documented tarsus-restricted expression profile (Fig 3A–3D) [56,57]. The other was divided into separate *TfAP-2*+ and *nub*+ domains. Based on our imaging, we propose that the *TfAP-2*+/*nub*− cluster corresponds to the ta1/ta2 joint, while the other corresponds to the tibia/ta1 joint. The tibia/ta1 joint can be further subdivided into the *nub*+ proximal and *TfAP-2*+ distal regions (Fig 3F; see S2A Fig for details of the high mt% cluster).

To understand the broader transcriptomic differences between joint regions, we tested for differential gene expression in each cluster compared to all other clusters in our joint dataset (a selection is given in Fig 3G). We recovered several genes with known roles in joint formation, including the *odd-skipped* family transcription factors *drm* and *sob* (Figs 3A, 3H, S2B and S2C). These were enriched at the interface between the proximal and distal tibia/ta1 clusters, consistent with their known absence from the upper tarsal segments [52]. We also looked at 2 further members of the same gene family: *odd*, which showed a similar expression pattern but was present in fewer cells, and *bowl*, which was widely expressed among the joint clusters and in the wider body of epithelial cells (S2D and S2E–S2G Fig). We observed 2 major patterns among the top differentially expressed genes (DEGs) defining the proximal region of the tibia/ta1 joint. First, genes that appeared widespread among epithelial cells, but which were excluded from the other joint clusters (e.g., *CG1648* and *Ser*; S2H and S2I Fig). Second, and more commonly, there were genes that showed strong specificity to the proximal tibia/ta1 joint (e.g., *Lim1* –Fig 3I–*nub*, *trh*, *caup*, *ara*, *pdm2*, *CG2016*, and *svp*; S2J–S2P Fig). To the best of our knowledge, no role for *pdm2* in leg development has been characterized, but it is believed to have evolved via duplication of *nub*, to which it is adjacent in the genome [58]. Our

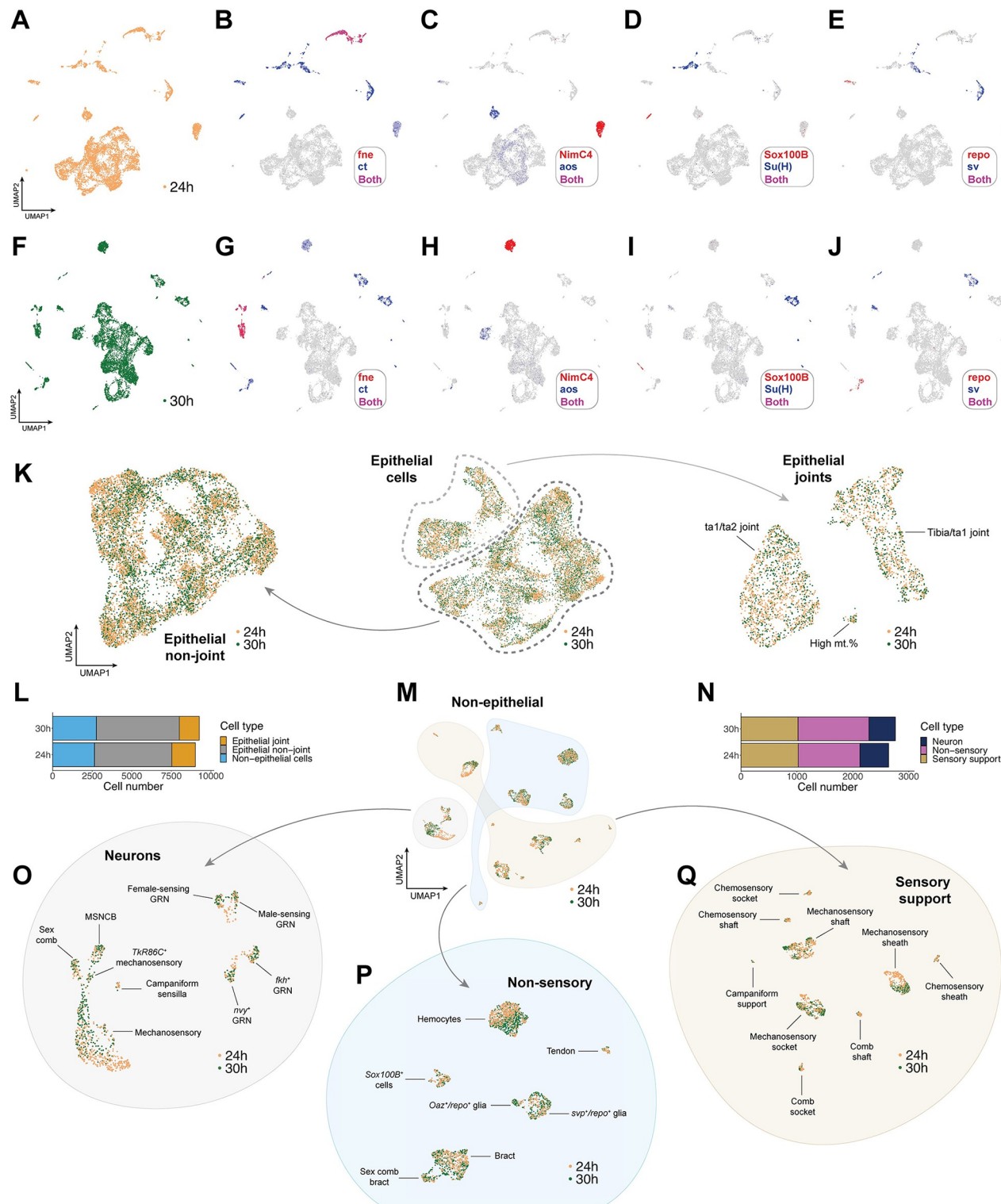

**Fig 2. Clustering and iterative subsetting of integrated 24 h and 30 h APF scRNA-seq datasets identifies tarsal cell types with high resolution.**
(A-J) UMAP plots showing cell clustering in the 24 h (A-E) and 30 h (F-J) datasets separately. Expression of a series of cluster markers is overlaid on the full 24 h (B-E) and 30 h (G-J) dataset UMAPs. At this resolution, each higher-level cluster in the 24 h dataset has a clear homolog in the 30 h dataset based on a selected subset of marker genes and vice versa. *fne* for neurons; *ct* for nonepithelial cells; *NimC4* for hemocytes; *aos* for bracts; *Sox100B* and *repo* for different subtypes of glia and axon-associated cells; *Su(H)* for socket cells; *sv* for shafts and sheaths. (K) The central UMAP

shows the clustering pattern observed in an integrated dataset containing just the epithelial joint and nonjoint cells from both 24 h (gold dots) and 30 h (green dots) samples. Joints are circled with a dashed light gray line, nonjoints with a dashed dark gray line. Reclustering the nonjoint and joint cells gave rise to the 2 flanking UMAP plots. See Fig 3 for details on how the annotations were determined. (L) The number of cells in the postfiltration, doublet-removed epithelial joint (yellow), epithelial nonjoint (gray), and nonepithelial cell (blue) datasets, plotted separately based on which sample (24 h APF or 30 h APF) the cells originated from. Numerical data with cell barcodes are listed in S1 Data. (M) UMAP showing the clustering pattern observed in an integrated dataset containing all nonepithelial cells from both the 24 h (gold dots) and 30 h (green dots) samples. Three major subsets of cells are grouped by colored shapes: neurons, nonsensory cells, and sensory support cells. (N) The number of cells in the postfiltration, doublet-removed sensory support (gold), neuron (navy), and nonsensory (pink) datasets, plotted separately based on which sample the cells originated from (24 h APF or 30 h APF). Numerical data with cell barcodes are listed in S2 Data. (O-Q) UMAPs showing the clustering pattern observed in integrated datasets containing all neurons (O), nonsensory cells (P), and sensory support cells (Q) from both the 24 h (gold dots) and 30 h (green dots) samples. See Figs 4–8 for details on how the annotations were determined. GRN, gustatory receptor neuron; MSNCB, mechanosensory neuron in chemosensory bristle. Data and code for generating the figure are available at https://www.osf.io/ba8tf.

data suggest that the two are coexpressed, a conclusion that's further supported by the recent discovery that the 2 genes share enhancers in the wing [59]. In contrast to the distinctiveness of the proximal tibia/ta1 expression profile, there was greater overlap in the DEGs for the distal tibia/ta1 and ta1/ta2 joints, with fewer cluster specific genes (Fig 3G; e.g., *fj*, Fig 3I; for UMAPs and discussion, see S2Q–S2V and S3A–S3R Figs). This suggests that while the transcriptomic distinctiveness of the proximal tibia/ta1 region is dominated by qualitative differences in expression, the more distal joint clusters show a more quantitative signal.

## Epithelial cells express a signature of anatomical position

We next turned to the largest portion of our dataset, the nonjoint epithelial cells. Here, we observed clear separation between dorsal and ventral cells. This separation was clearest when mapping the expression of *H15* (ventral) and *bi* (dorsal) (Fig 3J). *wg* (ventral) and *dpp* (dorsal) showed a similar, albeit weaker separation (Fig 3K). Unlike the joints, anterior–posterior separation was also clear, as delineated by the expression of *ci* (anterior) and *hh* (posterior) (Fig 3L). A weaker signature of proximal–distal separation could also be discerned from the expression of *bab2*, which is absent from the tibia and increases in expression between ta1 and ta2 (reviewed in [60]) (Fig 3M). We also detected localized expression of *rn*, the expression of which in ta1 is limited to the distal region [61] (Fig 3M). Recovery of spatial patterning in epithelial cells has recently been demonstrated in *Drosophila* wing imaginal disc scRNA-seq data [62,63]. But unlike in wing discs, we find that in this region of the leg the anterior–posterior signature is stronger than the proximal–distal, which may be due to our sequencing only a small fraction of the proximal–distal axis (i.e., just one tarsal segment). We then used the intersecting axes of positional marker expression as a spatial reference map to assign clusters to regions of the dissected leg tissue (Fig 3N and 3O). We tested for DEGs by comparing each region to the remainder (Fig 3P). Many DEGs showed signatures of localized up-regulation rather than cluster-specific expression, as would reasonably be expected from a tissue composed of a single cell type (exceptions include *lbl*, *CG13064*, *CG13065*, and *CG13046*; S3S–S3V Fig). But the cluster enriched for the distal ta1 marker *rn* exhibited a more specific gene expression profile, including showing enriched expression of the effector of sex determination *dsx* (reviewed in [64]), consistent with the localization of the sex comb to this region [65,66]. Genes enriched here represent candidate components of the sex-specific gene regulatory network that drives sex comb rotation (S3W–S3AE Fig).

## Pupal leg hemocytes form a uniform population

We identified a single hemocyte cluster based on enriched expression of *He*, *Hml*, *srp*, and Nimrod-type receptor genes (Figs 4C, 4E and S4A–S4N) [67–69]. In our differential gene expression analysis, we observed strongly hemocyte-enriched expression of genes including

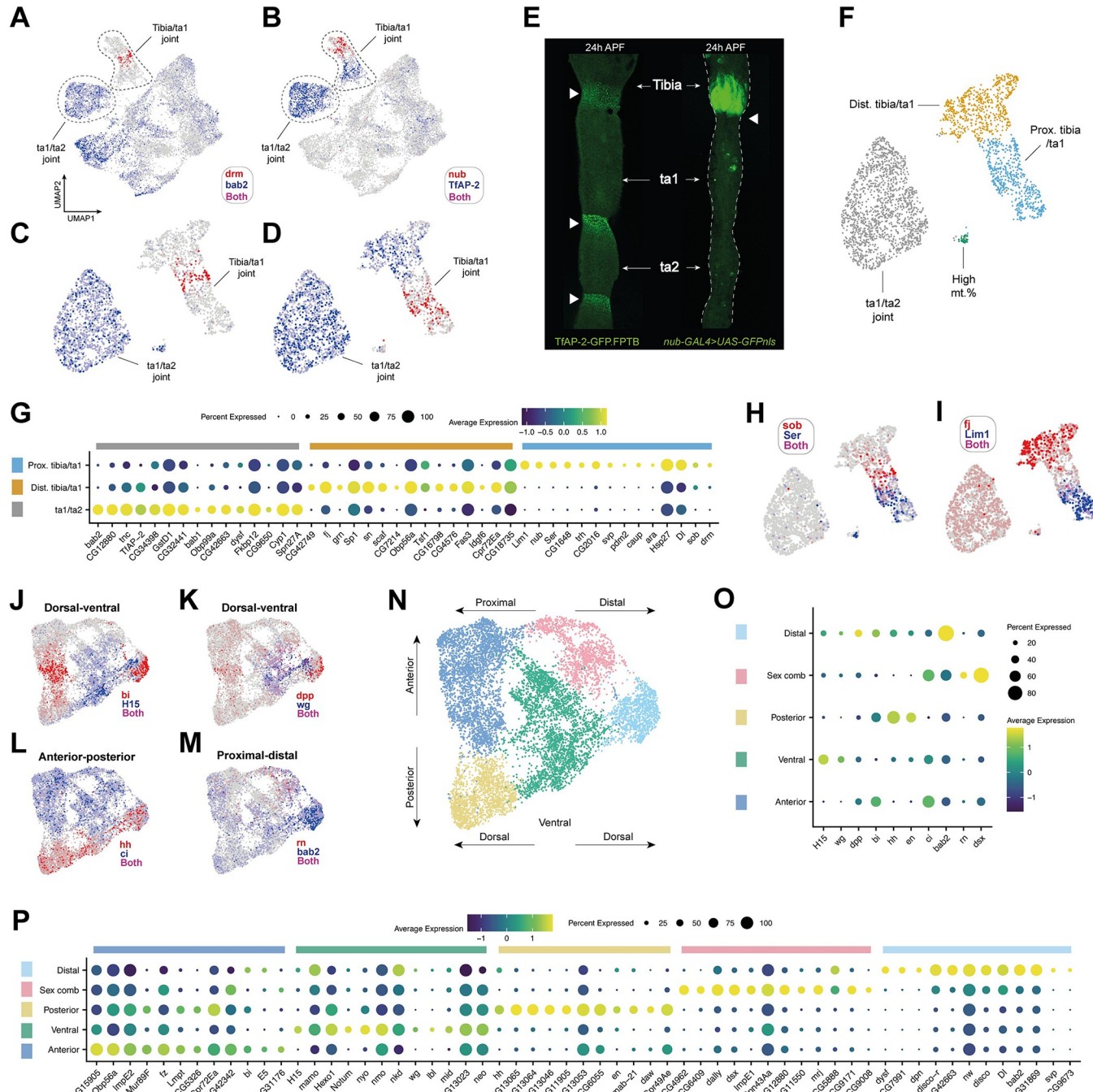

**Fig 3. Single-cell sequencing recovers positional information in the leg epithelium.** (A, B) UMAP plots of the integrated epithelial joint and nonjoint dataset overlaid with the expression of (A) *drm* (red) and *bab2* (blue), and (B) *nub* (red) and *TfAP-2* (blue). The 2 major joint clusters are circled and can be distinguished from one another based on the expression of these 4 genes. *bab* is known to be restricted to the tarsus starting with the distal first tarsal segment [56,57], providing one means through which to distinguish between the tibia and intertarsal joints. (C, D) As in (A) and (B) but with a UMAP plot of just the joint cells. (E) Confocal images showing 24 h pupal legs. On the left is a leg from a *TfAP-2-GFP* male. Staining is concentrated at the joints, which are each marked with a white triangle. Staining appears stronger at the intertarsal joints compared to the tibia/ta1 joint, consistent with the expression pattern of *TfAP-2* in the scRNA-seq data. On the right is a leg from a *nub-GAL4 > UAS-GFP.nls* male. Staining is concentrated in the distal tibia, proximal to the tibia/ta1 joint (marked with a white arrow). Note that some nonspecific staining from contaminating fat body is present in ta1 and ta2. (F) Joint UMAP with clusters identified through shared nearest neighbor clustering and annotated based on the data presented in (A-E). For details of the high mt. % cluster, see S2A Fig. (G) Dot plot of a selection of top marker genes for each of the joint clusters given in F (excluding the high mt. % cluster). Marker genes were identified by comparing each cluster to the remaining joint clusters. Dot size reflects the number of cells in the cluster in which a transcript for the marker gene was detected, while color represents the expression level. (H, I) Expression of a selection of top marker genes identified in the analysis presented in G overlaid on

the joint UMAP plot. (H) *sob* (red) and *Ser* (blue). (I) *fj* (red) and *Lim1* (blue). (J-M) UMAP plots of the nonjoint epithelial cells overlaid with markers of spatial identity. Both (J) and (K) show expression of dorsal (red: *bi* and *dpp*) and ventral (blue: *H15* and *wg*) markers. (L) shows anterior (blue: *ci*) and posterior (red: *hh*) markers. (M) shows proximal (blue: *bab2*) and distal (red: *rn*) markers. As is clear from the expression patterns, separation based on spatial markers is apparent for each axis, although stronger for dorsal–ventral and anterior–posterior than proximal–distal. This is likely due to us recovering only a small fraction of the proximal–distal axis by focusing in on just a single tarsal segment. (N) UMAP plot of the nonjoint epithelial cells colored by cluster identity as determined through shared nearest neighbor clustering. Spatial axes are illustrated by arrows based on the expression data presented in (J-M). (O) A dot plot showing the expression of positional markers across each cluster given in (N). Clusters are assigned to regions based on the positional gene expression signature they display. (P) A dot plot of the top markers for each cluster given in (N, O). Marker genes were identified by comparing each cluster to the remaining nonjoint epithelial clusters. Data and code for generating the scRNA-seq elements of this figure are available at https://www.osf.io/ba8tf.

*CG31777*, *CG31337*, *CG3961*, *CG14629*, *CG4250*, *Mec2*, *CG42369*, *CG5958*, *Glt*, and *CG10621* (Fig 4B; see S4X–S4AG Fig for UMAPs of the first 5 genes). To test for the presence of hemocyte subtypes, we ran a wide panel of recently identified lamellocyte, crystal cell, and plasmatocyte subtype-specific genes against our data [70]. However, we saw no obvious subclustering in relation to these genes: They were either widely expressed among hemocyte cells, too patchily expressed to reflect a clear subpopulation, or absent from our dataset (S4O–S4W Fig). The

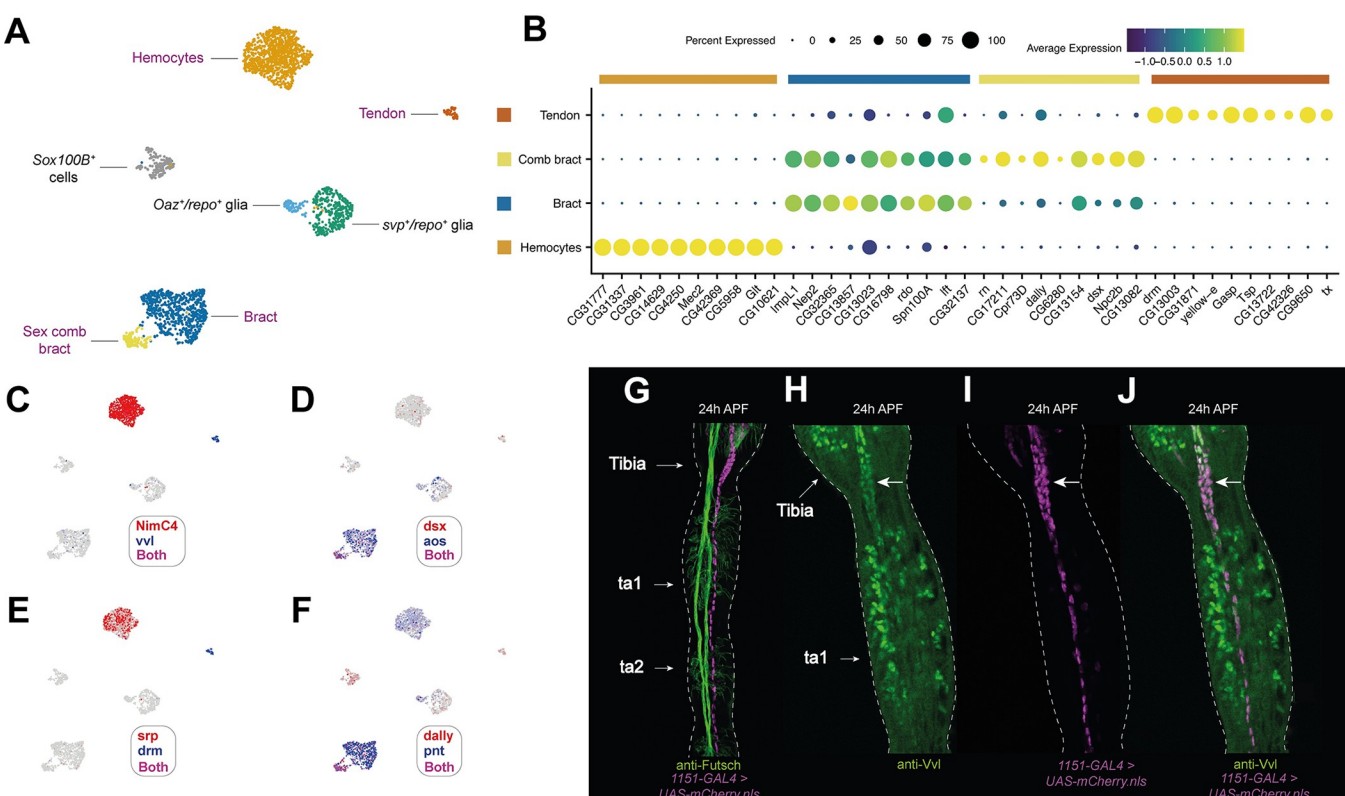

**Fig 4. The tarsus contains several types of nonsensory, nonepithelial cells.** (A) Annotated UMAP plot of nonsensory cells. Cell labels provided in purple indicate populations that are discussed in this figure. Those in black are discussed in Fig 5. (B) Dot plot of the expression of top differentially expressed genes identified through comparisons between each named cluster and all remaining clusters in (A). (C-F) The nonsensory UMAP shown in (A) overlaid with expression of key marker genes for each cluster. Note that *dsx* (D) and *dally* (F) are expressed in a distinct subset of bract cells, which likely corresponds to sex comb bracts. (G) 24 h APF male pupal upper tarsal segments showing staining from *1151-GAL4 > UAS-mCherry.nls* (magenta) and the neuronal marker anti-Futsch (green). *1151-GAL4* marks tendons [84]. The arrangement of tendon cells is clearly distinct from the paired nerve fibers that run along the same axis. (H-J) 24 h APF first tarsal segment and distal tibia from an *1151-GAL4 > UAS-mCherry.nls* (magenta) male counterstained with anti-Vvl (green). Costaining is clearer in the levator and depressor tendons at the distal tibia/ta1 joint (marked with an arrow) than in the long tendon, which extends along the proximal–distal axis of the tarsal segments. This may be due to the greater concentration of tendon cells in this region and difficulties distinguishing between anti-Vvl staining in mechanosensory bristle cells (see Fig 6Q–6S) and tendon cells. Data and code for generating the scRNA-seq elements of this figure are available at https://www.osf.io/ba8tf.

absence of clear subclustering may reflect the rarity of these hemocyte subpopulations, that recovered cells are insufficiently differentiated at these time points to discriminate subclasses, that these notably fragile cells [71] lyse during tissue dissociation, or point to differences between the larval and pupal immune cell repertoire, for which there is some evidence [72,73].

## The induction of bract identity is accompanied by a transcriptomic shift away from epithelial cell profiles

Most, but not all, mechanosensory bristles on the legs are associated with a bract cell [9,49]. We identified bracts based on the expression of the transcription factor *pnt* and *aos*, an EGF inhibitor selectively expressed in cells assuming bract fate [74–76]. Among our nonsensory clusters, we find 2 enriched for *aos* and *pnt* (Figs 4D, 4F, and S5A–S5D). Differential gene expression analysis comparing among our nonsensory cell clusters showed that the top markers of the major bract cluster also showed elevated expression in the minority cluster but that the top markers of the minority cluster were more specific in their expression (Fig 4B). The top DEGs for the minority cluster included several genes that were among the top markers of the putative sex comb bearing region identified in our epithelial cell analysis (*rn*, *dsx*, and *dally*; Fig 4D and 4F). Thus, these cells likely correspond to sex comb bracts. Given that bract identity is induced in epithelial cells, rather than emerging through the sensory organ precursor lineage, the natural comparison to make to identify putative determinants of bract fate is to compare bract cells with other epithelial cells [12,74,77–80]. Comparing the 2 bract clusters with the nonjoint epithelial cells, we observed several genes that were highly enriched in bracts and largely absent from epithelial cells (S5E Fig). It was common to find expression of some of these genes (e.g., *CG33110*, *Nep2*, *CG32365*, *neur*; S5F–S5P Fig) in bristle shaft and socket cells, which, considering the short bristle hair-like protrusion that bracts develop, may reflect their partially overlapping morphological characteristics. Ultimately, the distinct expression profile of the bracts suggests that induction of this identity in an epithelial cell is followed by remodeling of its transcriptome.

## Tendon cells express the POU transcription factor *vvl*, but not *sr*, between 24 h and 30 h APF

We identified a cluster of tendon cells based on the expression of *Tsp*, *tx*, and the joint marker *drm* (Fig 4B and 4E) [81–83]. To visualize the anatomical distribution of tendon cells in the focal leg region, we crossed the verified tendon marker line *1151-GAL4* [84] to *UAS-mCherry. nls* and counterstained with an antibody against the neuronal marker Futsch to distinguish between tendons and axonal trunks (Fig 4G). The dissected region contains the "long tendon," which runs along the proximal–distal axis of the tarsal segments, as well as the distal portion of the "tarsus levator" and "tarsus depressor" tendons, which are housed in the distal tibia [84]. One of the top DEGs for our $Tsp^+/tx^+$ cells was the POU homeobox transcription factor *vvl* (Fig 4C). When counterstaining *1151-GAL4>UAS-mCherry.nls* legs with an antibody raised against Vvl, we observed clear costaining, supporting a tendon cell identity for this cluster (Fig 4H–4J). Of the top DEGs we identified for tendon cells, none were entirely specific to this cluster when looking across all cells in the dataset. A common pattern was to see localized expression among epithelial cells (e.g., *drm*, *Tsp*, *CG13003*) or among sockets and shaft cells (e.g., *tx* and *CG42326*) (S6A–S6R Fig). Beyond the top 10 tendon cell DEGs, we detected significantly enriched expression of *trol*, which encodes the extracellular matrix proteoglycan Perlecan (validated using *trol-GAL4*; S6S–S6V Fig). Surprisingly, we did not detect enriched expression of the tendon-specifying transcription factor *sr* in this cluster, nor its tendon-specific downstream targets *slow* and *Lrt* [85,86] (S6W–S6AE Fig). Their absence may reflect the

developmental time points that we sequenced or differences between this tibia/tarsus tendon population and more commonly studied populations in the embryo [87] and wings [88].

## Glial cells in the developing first tarsal segment have noncanonical expression profiles

Three major classes of glia have been described in the leg: perineural and subperineural glia (which collectively comprise the surface glia), and the PNS-specific wrapping glia [89]. To identify these populations in our dataset, we mapped the expression of the canonical glia marker *repo*, which is thought to be expressed in all lateral/embryonic glial cells except for a subset of specialized wrapping glia in the CNS known as midline glia [90–94]. We observed 2 *repo*+ clusters, one of which was *svp*+ and the other *Oaz*+ (Fig 5A–5E). Previous work has shown that Oaz specifically labels wrapping glia in larval peripheral nerves, where it is coexpressed with Repo [95]. In our stainings, we observed that anti-Oaz labeled a small number of *repo-GAL4*+ cells (Fig 5N–5P). Beyond *Oaz*, however, the expression patterns of the *repo*+ cells were atypical with respect to known marker genes. The subperineural glia marker genes *moody* and *Gli* were widely expressed across *repo*+ cells, including in the *Oaz*+ putative wrapping glia cells (Fig 5H and 5I) [89,96]. Conversely, the wrapping glia markers *nrv2* and *Ntan1* weren't restricted to the *Oaz*+ cluster (Fig 5J and 5K), while the perineural glia marker *Jupiter* was widely detected across all nonsensory cells (S8F Fig). Indeed, genes previously shown to be enriched in surface glia [97] were similarly represented in the DEGs of each cluster (*svp*+ glia: 20/285, approximately 7.0%; *Oaz*+ glia: 14/223, approximately 6.3%; top DEGs are given in Fig 5B, with an expanded list in S7A Fig).

Despite these surprisingly broad expression patterns of established marker genes, 2 features point to known identities. First, *svp*-lacZ is known to be expressed in all surface glia [91]; among nonsensory cells in our data, *svp* was restricted to the *Oaz*−/*repo*+ cells (Fig 5D). Second, among *repo*+ cells, the surface glia markers *apt* and *Gs2* [89] appeared similarly restricted to the *Oaz*− cells (Figs 5L and S8G). Collectively, this suggests that our *Oaz*+ population correspond to wrapping glia, while the *svp*+ population corresponds to surface glia. This latter population appears heterogeneous, with localized up-regulation of *Gli* detectable (Fig 5I). Several processes may underlie this heterogeneity. One possibility is that it reflects developmental staging differences among surface glia. This is supported by the localized expression of the upstream determinant of glial identity, *gcm*, which is known to act early in development (Fig 5M) [98]. But another possibility is that this cluster includes a mix of both perineural and subperineural glia. Perineural glia are far more numerous than subperineural glia in the leg [89], so any we recover may present as a subregion within a surface glia cluster otherwise dominated by perineural glia. Finally, surface glia may naturally be heterogenous in their expression profiles, as has been suggested by others [89].

## A novel cell type associated with the neural lamella

Beyond the *repo*+ clusters, we observed a *repo*− population that was enriched for expression of the transcription factors *Sox100B* and *Lim1* (Fig 5F and 5G) and expressed the midline glia marker *wrapper* [99–101] (Fig 5AB). To verify the mutually exclusive expression patterns of *repo* and *Sox100B*, we counterstained *repo-GAL4 > UAS-mCherry.nls* legs with an anti-Sox100B antibody across multiple time points. First, we looked at 0 h APF leg discs (Fig 5Q and 5R). Here, we observed no overlap in expression but noted a difference in the behavior of these 2 cell populations: While *repo-GAL4*+ cells could be seen migrating into the leg disc, as CNS-derived glia are known to do [89], we detected no such migratory behavior in the Sox100B+ cells. To determine whether Sox100B+ cells migrate into the disc at an earlier time

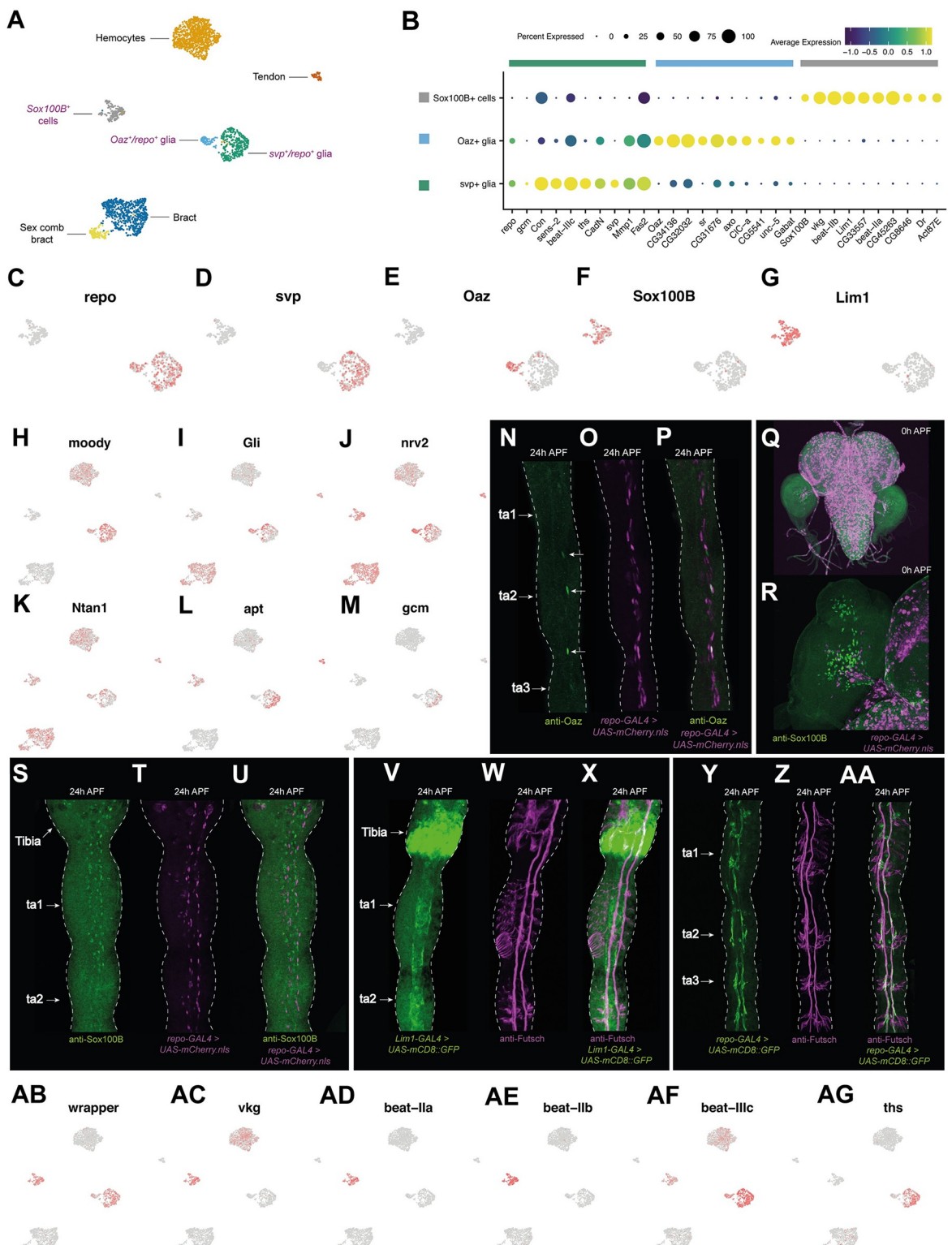

**Fig 5. Noncanonical expression patterns in leg glia and a new cell type associated with the neural lamella.** (A) Annotated UMAP plot of nonsensory cells. While the *Oaz*+ and *svp*+ glia cells express the canonical glia marker *repo*, the *Sox100B*+ cells do not. Cell labels provided in purple indicate populations that are discussed in this figure. Those in black are discussed in Fig 4. (B) Dot plot of the expression of *repo* and *gcm*, the canonical glia markers, along with top differentially expressed genes identified through comparisons between each of *Sox100B*+ glia, *Oaz*+/*repo*+ glia, and *svp*+/*repo*+ glia against all the clusters named in (A). Of these genes, *sr* is known to induce tendon cell fate, and to the

best of our knowledge, no functions have previously been reported for *sr* in glia. *Unc-5* and *Fas2* are both required for glial migration (reviewed in [93]). Genes identified as differentially expressed through more targeted between-glia comparisons are given in S7A Fig. (C-G) UMAP plots of the subsetted *Sox100B*+ cells, *Oaz*+/*repo*+ glia, and *svp*+/*repo*+ glia from (A) overlaid with the expression of a series of top marker genes for each cluster. (H-M) The UMAP plot shown in (A) overlaid with the expression of a series of glia markers. *moody* and *Gli* are subpineural glia markers, *nrv2* and *Ntan1* are wrapping glia markers, *apt* is a surface glia marker (i.e., a marker of both perineural and subpineural glia), and *gcm* is the upstream determinant of glial identity [89,96]. (N-P) 24 h APF legs from *repo-GAL4>UAS-mCherry.nls* males counterstained with anti-Oaz, a marker of wrapping glia [95]. Oaz+ cells are denoted by an arrow in the left-hand image. In Oaz+ cells at 24 h APF, the *repo-GAL4*+ signal was often weak and in one of the 4 legs we imaged, the one shown here, we observed a single *Oaz*+ cell that appeared *repo-GAL4*−. This is the topmost of the Oaz+ cells to which an arrow is pointing. (Q, R) Brain, ventral nerve cord, and leg discs (Q) and a close-up of a leg disc (R) from *repo-GAL4>UAS-mCherry.nls* males counterstained with anti-Sox100B. Note how *repo-GAL4*+ cells can be seen migrating into the disc from the CNS, while Sox100B+ cells appear to originate within the disc itself. (S-U) 24 h APF male upper tarsal segments from *repo-GAL4 > UAS-mCherry.nls* counterstained with anti-Sox100B. Both show a similar, but nonoverlapping, distribution of stained cells. (V-X) 24 h APF male upper tarsal segments from *Lim1-GAL4 > UAS-mCD8::GFP* counterstained with anti-Futsch. Above the tibia/ta1 joint, *Lim1-GAL4* was expressed in the epithelial cells of the distal tibia, as predicted by our epithelial joint analysis (Fig 3I). Below the joint, the staining surrounded and spanned the distance between the 2 central axon trunks into which the sensory neuron axons project. (Y-AA) 24 h APF male upper tarsal segments from *repo-GAL4 > UAS-mCD8::GFP* counterstained with anti-Futsch. Unlike the *Lim1-GAL4*, *repo-GAL4* staining does not span the gap between the 2 axon trunks with which it is closely associated, and cell bodies are clearly seen branching away from the fibers. (AB-AG) The UMAP plot shown in (A) overlaid with the expression of a series of top markers identified in this study. Data and code for generating the scRNA-seq elements of this figure are available at https://www.osf.io/ba8tf.

point or originate within the disc itself, we next looked at L2 and L3 larval discs (S7B and S7C Fig). We observed migration of *repo-GAL4*+ cells across both time points. In contrast, Sox100B+ cells were present in the discs only in L3 larvae and showed no clear indication of migration. We therefore conclude that Sox100B+ cells originate within the leg disc itself.

At 24 h APF, we again detected no overlap in *repo-GAL4* and anti-Sox100B staining in the legs. Despite being *repo-GAL4*−, the nuclei of Sox100B+ cells occupied glia-like positions in relation to the axon trunks at this time point (Fig 5S–5U). A glia-like association between Sox100B+ cells and the neuropil was also evident in the larval (S7B Fig) and pupal (S7D Fig) ventral nerve cord. In 24 h APF legs, the morphology and neuronal associations of Sox100B+ and *repo-GAL4*+ cells were clearly distinct. Expressing a membrane-tethered form of GFP under the control of a GAL4 driver for one of the top markers of the *Sox100B*+ cluster (Lim1-GAL4; [102]), we observed that these cells appear to surround the axon trunks (Fig 5V–5X). When compared to the equivalent staining for *repo-GAL4*, the staining around the axon trunks in *Lim1-GAL4 > UAS-mCD8::GFP* legs appeared larger in diameter, suggesting that it comprises a layer that is outer to that of the *repo-GAL4*+ cells (Figs 5Y–5AA, S8A, and S8B). To exclude the possibility that these cells correspond to myoblasts, we compared the distribution of Sox100B+ cells to those stained by an antibody raised against the myoblast marker Mef2 (S8C Fig) [103]. Unlike Sox100B+ cells, myoblasts were restricted to the tibia and absent from tarsal segments, arguing against a myoblast identity for Sox100B+ cells.

We next visualized a protein trap of one of the *Sox100B*+ cluster's most specifically enriched markers, *vkg*, which encodes a subunit of the extracellular matrix component Collagen IV and which was essentially absent from the *repo*+ glia (Figs 5AC and S8D). The vkg::GFP staining pattern resembled *Lim1-GAL4>UAS-mCD8::GFP*: a broad ensheathing of the axon trunks that extended more laterally than the equivalent staining observed for *repo-GAL4* (S8D Fig). This provides further support for the "outerness" of these cells. vkg::GFP is known to label the neural lamella, a dense network of extracellular matrix that surrounds the central and peripheral nervous systems and which is required to help control their shape [96,104,105]. These features suggest that the *Sox100B*+ cells we identify here may be involved in the construction of the neural lamella. This role is thought to be performed by migrating hemocytes during embryogenesis [96,106] and consistent with this we find that leg hemocytes also express *vkg*, albeit less strongly (Fig 5AC). But beyond this, the only other clear similarity we detect between our hemocyte and *Sox100B*+ cells was the expression of *NimC3*, which was highly

expressed in hemocytes and showed low-level expression in *Sox100B*⁺ cells (S4K Fig). It's therefore possible that migrating hemocytes work alongside *Sox100B*⁺ cells to construct the neural lamella. Consistent with this, we see both cluster-specific and overlapping expression of many extracellular matrix component genes: *SPARC* is expressed in *Sox100B*⁺ cells, hemocytes, and *repo*⁺ glia; *trol* shows low-level expression in *Sox100B*⁺ cells, hemocytes, and *svp*⁺/*repo*⁺ glia; *Col4a1* and *vkg* are present in both *Sox100B*⁺ cells and hemocytes; and *Pxn* is hemocyte-specific (Figs 5AC and S8H–S8K). However, the position of *Sox100B*⁺ cells at the outer layer of glia, along with their strong enrichment of the canonical neural lamella marker *vkg*, suggest that this cell type may play the primary role in neural lamella synthesis during pupal development, with any contribution by the hemocytes being secondary.

### *repo*⁺ glia and *Sox100B*⁺ cells express distinct cell–cell communication gene repertoires

Comparing between the *repo*⁺ glia and *Sox100B*⁺ cells, we observed cluster-specific expression of beaten path family genes. *beat-IIa* and *beat-IIb* were restricted to the *Sox100B*⁺ cluster, while *beat-IIIc* was enriched in *repo*⁺ cells (Fig 5AD and 5AF). *beaten path* genes are thought to act as neuronal receptors for sidestep gene family ligands expressed in peripheral tissues [107]. Thus, the differential expression of different subsets of beaten path family genes between *repo*⁺ and *Sox100B*⁺ cells point both to the importance of these genes in the nonneuronal component of the nervous system and to cell type–specific patterns of between-cell communication in the developing nervous system. These *beat* genes have been recorded elsewhere in the fly: In the visual system, *beat-IIb* is expressed in L3 and L4 lamina neurons, the glia beneath L5, and in the lamina neuropil, while *beat-IIIc* is expressed in a subset of retinal neurons [108]. Other cell communication pathway elements also showed cell type specificity. For example, we observed that the fibroblast growth factor (FGF) ligand *ths* was, among glia, largely restricted to *svp*⁺/*repo*⁺ cells (Fig 5AG). FGF signaling is known to underlie aspects of neuron–glia communication in *Drosophila*, although in these cases, the source of Ths is neuronal. In one example, Ths is thought to act in neurons as a directional chemoattractant for the migration of astrocytes and the outgrowth of their processes [109]. In 2 others, the release of Ths from olfactory neurons directs ensheathing glia to wrap each glomerulus [110], while Ths in photoreceptor neurons induces differentiation of glia in the developing eye [111]. The expression of *ths* in one of our glia populations, specifically the population we believe to correspond to the surface glia, is therefore surprising. It's possible that FGF-mediated interactions between different glia populations guide their concerted differentiation and the development of the close physical associations they form. Similar interactions with neurons may also help guide the growth of neuronal projections in the vicinity of these glia.

### A combinatorial transcription factor code for sensory neurons

We recovered multiple distinct sensory neuron populations in our clustering analysis, each defined by the expression of a unique combination of transcription factors (Fig 6A and 6B). We recovered a similar clustering pattern in our analysis of male neurons from the Fly Cell Atlas (FCA) adult leg data (Fig 6G and 6H) [38]. In contrast to our pupal data, however, the FCA dataset is derived from all segments of all 3 pairs of legs and is single-nuclei, rather than single-cell. We failed to recover clear sex comb or MSNCB (mechanosensory neuron in chemosensory bristle) populations in the FCA data. But in their place, we recovered 3 populations apparently absent from our pupal dataset. These novel clusters were enriched for *CG9650*, a transcription factor we found to be enriched in joint and tendon cells, and showed cluster-specific, combinatorial expression of transcription factors, including *erm* and *bab1* (Fig 6I and 6J).

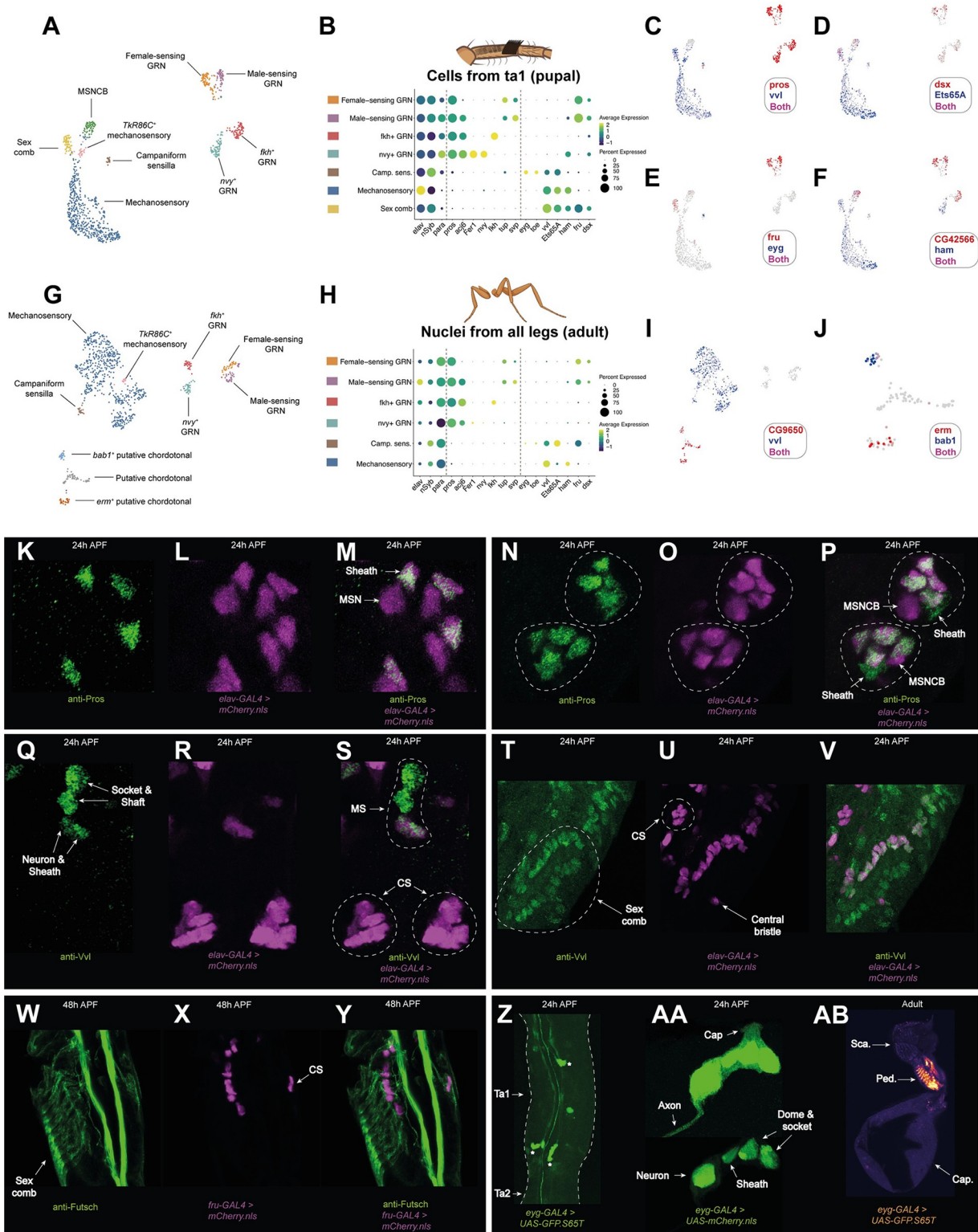

**Fig 6. Identification of a combinatorial transcription factor code for leg sensory neurons.** (A) Annotated UMAP plot of neuronal cells from the integrated 24 h AFP and 30 h APF first tarsal segment dataset. GRN, gustatory receptor neuron; MSNCB, mechanosensory neuron in chemosensory bristle. See S9A–S9H Fig for details on the *TkR86C*+ mechanosensory neurons. (B) Dot plot showing the expression of a series of canonical neuronal markers (*elav*, *nSyb*, and *para*) and transcription factors across the major neuron class clusters labeled in the UMAP given in (A). Each cluster expresses a unique combination. The dotted lines separate the canonical neuronal markers and then the chemoreceptor from

mechanoreceptor organs transcription factor markers. (C-F) UMAP plot of neuronal cells from the integrated 24 h AFP and 30 h APF first tarsal segment dataset overlaid with the expression of members of the transcription factor code depicted in (B). (C) Note how the MSNCB cluster branching off from the top of the mechanosensory neuron population is negative for both *vvl* and *pros*. (D) *Ets65A* is present in all non-GRN populations in the UMAP, while an effector of sex differentiation, *dsx*, is expressed in GRNs, sex comb neurons, and MSNCBs. (E) *fru*, the other effector of sex differentiation, is enriched in 2 GRN populations and sex comb neurons, while *eyg* is restricted to campaniform sensilla neurons. (F) *CG42566* is the only nontranscription factor plotted. It is a top marker of MSNCBs and its expression in both MSNCBs and GRNs contributed to this cluster's chemosensory bristle annotation. *ham* is enriched in mechanosensory neuron classes and 2 GRN populations. (G) Annotated UMAP plot of male neuronal cells subsetted from the Fly Cell Atlas single-nuclei RNA-seq leg dataset [38]. Note the presence of 3 clusters, annotated as "putative chordotonal," which are absent from the pupal dataset—chordotonal organs are not present in the upper tarsal segments. No clear MSNCB or sex comb clusters could be resolved in this dataset. (H) As (B) but for the male neuronal cells subsetted from the Fly Cell Atlas single-nuclei RNA-seq leg dataset. Only those clusters present in the pupal single-cell data are shown. (I) UMAP plot of male neuronal cells subsetted from the Fly Cell Atlas single-nuclei RNA-seq leg dataset overlaid with expression of the mechanosensory neuron marker *vvl* (blue) and a top marker of the putative chordotonal organs, the predicted transcription factor *CG9650* (red). (J) A subset of (I), showing only the putative chordotonal clusters overlaid with expression of 2 transcription factors, *bab1* (blue) and *erm* (red). (K-V) Confocal images of 24 h APF male first tarsal segments. (K-M) Mechanosensory bristles from *elav-GAL4 > UAS-mCherry.nls* (magenta) stained with anti-Pros (green). Two *elav-GAL4*$^+$ cells are present per mechanosensory bristle, one of which, the sheath, is Pros$^+$. *elav-GAL4* expression in the sheath is likely due to the legs being imaged soon after the division of the common pIIIb progenitor cell from which they derive (see also [113] and S9K–S9M Fig). MSN, mechanosensory neuron. (N-P) Two chemosensory bristles (circled) from *elav-GAL4 > UAS-mCherry.nls* (magenta) stained with anti-Pros (green). Note that each bristle includes 4 Pros$^+$/*elav-GAL4*$^+$ cells (the gustatory receptor neurons), 1 Pros$^+$/*elav-GAL4*$^-$ cell (the chemosensory sheath cell), and 1 Pros$^-$/*elav-GAL4*$^+$ cell (the MSNCB, mechanosensory neuron in chemosensory bristle). (Q-S) Two chemosensory (CS) bristles and 1 mechanosensory (MS) bristle from *elav-GAL4 > UAS-mCherry.nls* (magenta) stained with anti-Vvl (green). Note that anti-Vvl staining is entirely absent from the CS bristle including, therefore, the mechanosensory neuron (MSNCB) that innervates it. Conversely, anti-Vvl staining is observed in all 4 constituent cells of a MS bristle. (T-V) The same stainings performed in (Q-S) but centered on the sex comb. Anti-Vvl staining is present in both the neuronal (*elav-GAL4*$^+$) and nonneuronal cells of the sex comb. The "central bristle," which develops from the same bristle row as the sex comb is labeled. (W-Y) Confocal images of 48 h APF male first tarsal segments showing the expression of *fru-GAL4* (magenta) and anti-Futsch (green). *fru-GAL4* expression is restricted to the sex comb and chemosensory (CS) neurons. The later 48 h time point was used as *fru-GAL4* was undetectable up until 40 h and weak up until 48 h. (Z) Confocal image of the first tarsal segment from a 24 h male from *eyg-GAL4 > UAS-GFP.S65T*. Campaniform sensilla are marked with asterisks. The axonal projections can be seen as parallel lines running either side of the central autofluorescence. Note that some nonspecific fat body staining is also present in this image. (AA) Confocal image of a distal first tarsal segment campaniform sensillum from a 24 h male where *eyg-GAL4* is driving the expression of *UAS-mCherry.nls*. The top and bottom image in this panel show the same sensillum but with different levels of saturation to variously highlight the domed structure (top) and the individual cells of the organ (bottom). (AB) As (Z) but showing an adult haltere. Note that the staining is restricted to the campaniform sensilla field on the pedicel ("Ped.") and apparently absent from the field on the scabellum ("Sca."). Data and code for generating the scRNA-seq elements of this figure are available at https://www.osf.io/ba8tf.

Consistent with signs of a joint identity, we believe these are likely to correspond to chordotonal neuron populations, a class that is absent from the first tarsal segment. The pupal and FCA neuron datasets did not integrate as well as our pupal datasets did to each other (S9I and S9J Fig). In light of the significant differences in sample preparation, age, cells versus nuclei, and dissected region between the datasets we opted to analyze them separately.

## Unique combinations of the transcription factors *vvl*, *pros*, and *fru* distinguish between mechanosensory, chemosensory, and sex comb neurons

In both our pupal and the FCA data, the nonoverlapping expression of *vvl* and *pros* marked the highest order difference between neuron clusters (Fig 6B, 6C, and 6H). Among sensory organ cells, *pros* is known to be expressed in sheath cells [112] and has been detected in DA1, DL3, and VA1d olfactory receptor neurons in scRNA-seq data from antennae [47]. Consistent with sheath cell expression, we observed anti-Pros staining in 1 cell per mechanosensory bristle (Fig 6K). Surprisingly, however, in many cases the Pros$^+$ cell was also positive for the canonical neuron marker *elav* (Fig 6L and 6M). We generally observed *elav-GAL4* expression in 2 cells per bristle, despite mechanosensory bristles being singly innervated (Fig 6L and 6M). The expression of *elav-GAL4* in the mechanosensory sheath cell likely reflects the early time point (24 h APF) at which we imaged. This conclusion was also reached by Simon and colleagues [113] after detecting *elav* expression in sheaths in 28 h APF mechanosensory bristles on the notum. Consistent with this conclusion, we observed a heterogenous mix of Pros$^+$/*elav-GAL4*$^+$

and Pros$^+$/*elav-GAL4*$^-$ mechanosensory bristle sheaths in some legs, suggesting between-mechanosensory bristle variation in developmental stage (S9K–S9M Fig). Expression of *elav* beyond the neuron has now been noted several times: Its expression appears surprisingly broad and its neural specificity dependent on posttranscriptional repression outside of neurons [35,114].

Unlike in mechanosensory bristles, we detected multiple Pros$^+$ cells per chemosensory bristle at 24 h APF (Fig 6N–6P). One of these was *elav-GAL4*$^-$, consistent both with a sheath identity and, when contrasted against the heterogenous *elav-GAL4* expression we observed among mechanosensory sheaths, the developmental timing differences that are known to exist between mechanosensory and chemosensory bristles: Chemosensory bristles are specified earlier than all but the largest mechanosensory bristles [115,116]. However, in contrast to mechanosensory bristles, anti-Pros staining in chemosensory bristles was not restricted to the sheath, extending to a further 4 *elav-GAL4*$^+$ cells (Fig 6N–6P). Mirroring this arrangement, we detected 4 Pros$^+$ populations in both the pupal and FCA data, suggesting that these correspond to 4 distinct GRN subtypes. Not all *elav-GAL4*$^+$ cells in the chemosensory bristle were Pros$^+$ though: We observed a single Pros$^-$/*elav-GAL4*$^+$ cell per bristle, which likely corresponds to the MSNCB (see below). Surprisingly, we failed to detect *Poxn*, which is known to be both necessary and sufficient for a chemosensory rather than mechanosensory fate, in GRNs in either dataset, which suggests that *Poxn* transcription may have ceased by 24 h APF [117–121].

In mechanosensory bristles, anti-Vvl marked all 4 of the constituent cell types (socket, shaft, neuron, and sheath) (Fig 6Q–6S). In contrast, no cells in chemosensory bristles were marked by anti-Vvl. Consequently, the mechanosensory neurons innervating mechanosensory and chemosensory bristles can be distinguished based on the presence of Vvl in the former and not the latter. In our pupal scRNA-seq data, we observe a cluster of *vvl*$^-$ cells branching off from the major *vvl*$^+$ neuron population (Fig 6C). This cluster is otherwise positive for transcription factors present in the major mechanosensory population, such as *Ets65A* and *ham* (Fig 6D and 6F) and therefore likely corresponds to MSNCBs. Consistent with this, one of the top markers for the cluster, *CG42566*, was absent from the major mechanosensory population but expressed in 3 of the 4 GRN populations (Fig 6F). *CG42566* remains largely restricted to GRNs in the adult FCA data (S9N Fig). Similar to MSNCBs, we observed an additional cluster branching off from the major mechanosensory population in the pupal data. However, this population was *vvl*$^+$ and, uniquely among nonchemosensory neurons in the region, heavily enriched for *fru*. Our stainings suggest that these cells correspond to the sex comb neurons, which we observed to be both Vvl$^+$ and *fru-GAL4*$^+$ (Fig 6T–6Y; see also [122] for a previous report of *fru* expression in sex comb neurons). Collectively, these 3 transcription factors—*pros*, *fru*, and *vvl*—represent candidate high-level regulators of networks of downstream genes specific to the neurons of 3 major sensory organ classes: mechanosensory bristles, chemosensory bristles, and the sex comb.

## Campaniform sensilla express the Pax family transcription factors *eyg* and *toe*

In our pupal data, we resolved a small cluster of cells enriched for the Pax family transcription factors *eyg* and *toe*. Expressing *UAS-GFP* under the control of *eyg-GAL4*, we observed staining in the regions of the first tarsal segment that correspond to the positions of the campaniform sensilla: 1 proximal organ and 2 distal organs [8] (Fig 6Z). Repeating the experiment with a nuclear-localizing *UAS-mCherry*, we detected expression in 4 cells within each tarsal campaniform sensillum, which presumably correspond to the neuron, sheath, socket, and dome cell (Fig 6AA). Although the expression of *eyg* and *toe* were relatively low in the adult nuclei campaniform sensilla neuron cluster, we observed *eyg-GAL4* activity in both adult legs (S9O Fig)

and in the adult haltere (Fig 6AA and 6AB). In the haltere, *eyg-GAL4* activity was detectable in the field of campaniform sensilla on the pedicel, but not the scabellum, raising the possibility that there exist distinct subtypes of campaniform sensilla that express unique gene repertoires.

## The legs contain 4 gustatory receptor neuron (GRN) classes, each expressing a unique combination of the transcription factors *acj6*, *fru*, *nvy*, and *fkh*

Our stainings showed that among neurons Pros was restricted to chemosensory bristles, suggesting that the multiple *pros*+ neuron clusters we detect in our scRNA-seq data represent subclasses of GRNs. Each GRN cluster expressed a unique combination of 5 transcription factors: *pros*+/*acj6*+/*nvy*+, *pros*+/*acj6*+/*fkh*+, *pros*+/*acj6*+/*fru*+, and *pros*+/*fru*+ (Fig 7A–7F). We validated these combinations using *fru-GAL4* in conjunction with antibodies raised against Pros, Acj6, Nvy, and Fkh (Fig 7G–7AA). Multiple bristles are often closely associated within a single region of the leg, while the neurons themselves frequently overlap within a single bristle. Consequently, it wasn't possible to definitively determine whether 1 cell of each GRN class was present in every ta1 bristle, but from our observations, this seems likely to be the case. Consistent with this, the numbers of cells recovered in each cluster generally appeared similar (Fig 7A). We recovered the same 4 populations, marked by the same transcription factor code, in the FCA full leg dataset, again recovering a similar number of cells in each GRN population (Fig 7AC). The expression of *nvy* was, however, far less extensive than in the pupal data, suggesting that expression drops off during later pupal development or that *nvy* is restricted to a subset of cells in this GRN class. Correspondence between the *nvy*+ cluster in the pupal and adult data was supported by additional marker genes, such as *foxo* and *Fer1* (see below). Among neurons, the 4 GRN populations showed specific or enriched expression of *Ir25a*, *Ir40a*, *Gluclalpha*, *RhoGAP102A*, *Snmp2*, *CG42540*, *CG13578*, *Tsp47F*, and *CG34342* (S10A–S10T Fig). Taken together, the recovery of the same 4 GRN classes across different time points, technologies, and dissected regions suggests that despite substantial between-bristle variation in receptor expression and sensitivity to given stimuli [19], just 4 core GRN classes might be present in the leg and that these classes are defined by combinatorial expression of a small set of transcription factors (Fig 7AB).

## Male- and female-sensing GRNs express distinct receptor repertoires

Previous work has shown that there are 2 *fru*+ GRNs per leg chemosensory bristle [21,22]. Both express the ion channels *ppk23* and *ppk29*, while one additionally expresses *ppk25* and *VGlut* [21,22,123–126]. These *ppk25*+/*VGlut*+ neurons respond to female pheromones, while the *ppk25*−/*VGlut*− cells respond to male pheromones [22]. In the pupal data, we observe *ppk23* in both *fru*+ populations, with a small number of *ppk29*+ cells also distributed across both (S10U and S10V Fig). Although *ppk25* was absent in our dataset, we observed a sharp divide between *fru*+/*acj6*+ and *fru*+/*acj6*− cells based on *VGlut* expression (S10W Fig). *VGlut* is restricted to the *fru*+/*acj6*− population, identifying these as the female-sensing neurons and the *fru*+/*acj6*+ cells as male-sensing. Receptor expression was more readily detected in the adult nuclei data, suggesting that receptors are generally expressed later in development than the time points we sequenced. In the adult dataset, we again observed that *VGlut* was restricted to the *fru*+/*acj6*− population and recovered the previously documented expression patterns of *ppk23*, *ppk25*, and *ppk29* (S10X–S10AA Fig). However, we also observed GRN-specific expression patterns of several other *ppk* genes. While *ppk25* was restricted to female-sensing cells, *ppk10* and the less frequently detected *ppk15* were restricted to male-sensing cells (S10AB–S10AD Fig). These therefore represent candidate male pheromone receptors. Conversely, *Ir21a* and *CG46448* appeared to be largely restricted to female-sensing neurons (S10AE–

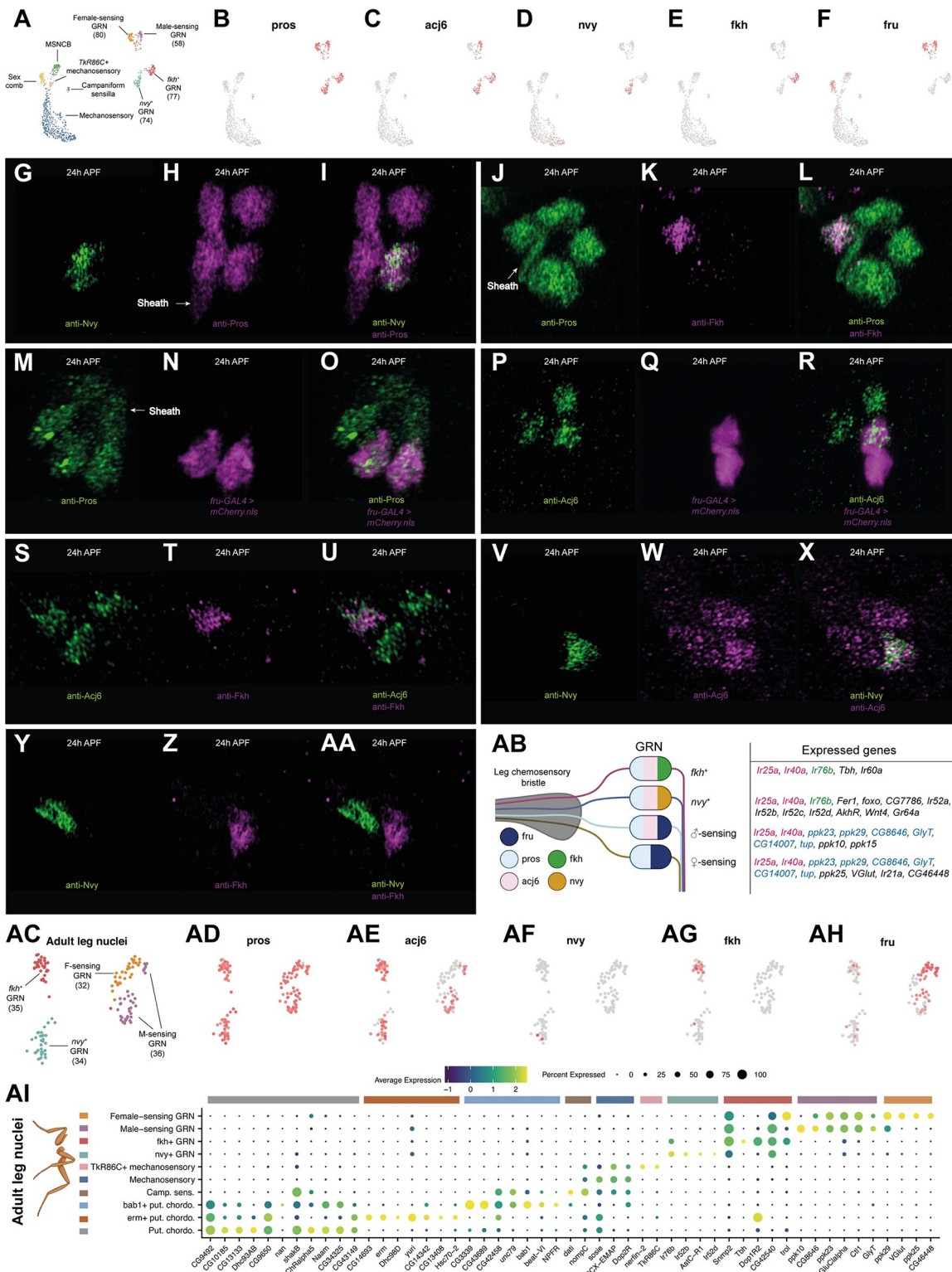

**Fig 7. Four gustatory receptor neuron (GRN) classes express a combinatorial transcription factor code and unique gene repertoires.** (A) Annotated UMAP of the pupal integrated neuron data. GRN, gustatory receptor neuron; MSNCB, mechanosensory neuron in chemosensory bristle. The number of cells in each GRN cluster is presented. The numbers are generally similar between each GRN population, with the exception of the *fru*+ male-sensing GRNs. This population was closely associated with the *fru*+ female-sensing GRNs, more so than were any other 2 GRN subtypes, and the interface between them in UMAP space contained several cells bearing

intermediate characteristics. Consequently, the discrepancy in cell numbers between *fru*+ GRN populations may reflect classification errors due to transcriptomic similarities. (B-F) The UMAP shown in (A) overlaid with the expression of 5 transcription factors (*pros*, *acj6*, *nvy*, *fkh*, and *fru*) that are expressed in unique combinations in each of the 4 GRN clusters. (G-AA) Testing the GRN transcription factor code derived from the scRNA-seq data on 24 h APF male first tarsal segments. (G-I) anti-Nvy (green) and anti-Pros (magenta). Note that 1 Pros+ cell is partially obscuring the Pros+ sheath cell, which has a distinct, elongated morphology. (J-L) Anti-Pros (green) and anti-Fkh (magenta). (M-O) Anti-Pros (green) and *fru-GAL4 > UAS-mCherry.nls* (magenta). (P-R) Anti-Acj6 (green) and *fru-GAL4 > UAS-mCherry.nls* (magenta). (S-U) Anti-Acj6 (green) and anti-Fkh (magenta). (V-X) Anti-Nvy (green) and anti-Acj6 (magenta). (Y-AA) Anti-Nvy (green) and anti-Fkh (magenta). (AB) A schematic summarizing the expression patterns of each transcription factor across GRNs, along with a selection of other genes detected in each subtype. Gene names are colored pink, green, or blue when shared across multiple GRN subtypes. (AC-AH) Recovery of the same transcription factor code in the Fly Cell Atlas single-nuclei adult male leg neuron data. Note that in the adult data, *nvy* is barely detected. Correspondence between the *nvy*+ cluster in the pupal and adult data was supported by additional marker genes, such as *foxo* and *Fer1* (see S11B, S11C, S11H, and S11I Fig). (AC) As in (A), the number of cells in each GRN population is presented. In this dataset, a subregion of what unsupervised clustering labeled as the *fru*+/*acj6*− population showed *acj6* expression, suggestive of a classification error. This conclusion is further supported by the *VGlut*, *ppk25*, and *ppk10* data below (see S10Y–S10AB Fig). We therefore manually labeled these as part of the *fru*+/*acj6*+ cluster. As in the pupal data, the interface between the 2 *fru*+ populations appeared particularly close. (AI) A dot plot summarizing the expression of a selection of top differentially expressed genes for each cluster that we identified in the Fly Cell Atlas single-nuclei adult male leg neuron data. Data and code for generating the scRNA-seq elements of this figure are available at https://www.osf.io/ba8tf.

S10AF Fig). *CG46448* is adjacent to *VGlut* on chromosome 2L, so their apparent coexpression in female-sensing neurons points to the possibility of their transcriptional control by shared *cis*-regulatory elements. These genes aside, the expression differences we observed between male- and female-sensing neurons were generally limited, but several genes were common to both pheromone-sensing populations, including the transcription factors *tup* and *svp*, the predicted sulfuric ester hydrolase *CG8646*, *CG14007*, and the glycine transporter *GlyT*, the latter suggesting that these neurons are glycinergic [127] (S10AG–S10AK Fig).

## *nvy*+ GRNs correspond to a sexually dimorphic population required for normal mating behaviour

The *nvy*+ GRNs showed specific expression of a further set of transcription factors in the pupal data, namely *Fer1*, *foxo*, and *CG7786* (S11A–S11D Fig). At these time points, few other genes showed enrichment in the *nvy*+ GRN cluster—exceptions include the phosphodiesterase *Pde6* and G-protein–coupled receptor *TrissinR* (S11E and S11F Fig). To further probe the identity of these neurons, we turned to the adult dataset. The transcription factor code identified in the pupal data largely persisted in the adult data, although each was detected in fewer cells and *CG7786* was absent (Figs 7AF and S11G–S11J). *AkhR* and *Gr64a* were among the top markers of the *nvy*+ GRNs in the adult data (S11M and S11N Fig). But like the other 5 *Gr* genes we detected across the 2 datasets, *Gr64a* was present in only a handful of cells (S11O–S11T Fig). The sparse detection of *Grs* may stem from gene dropout due to low abundance or reflect their restricted expression among GRNs of the same class. *Ir52a*, *Ir52b*, *Ir52c*, and *Ir52d* were also among the top markers of the *nvy*+ GRN cluster (S11U–S11X Fig). With the exception of *Ir52b*, these ionotropic receptor (IR) genes have been well characterized: They are known to be largely coexpressed in a subset of leg taste bristles enriched in the first tarsal segment, to show quantitative differences in expression between males and females, to show sexual dimorphism in their projections (they cross the midline in a commissure in males), to be required for normal sexual behavior, and to be expressed in neurons distinct from those involved in sweet or bitter sensing [20,128]. *Ir52c* and *Ir52d* are further known to be restricted to the forelegs [20]. Despite their sexually dimorphic characteristics, the neurons expressing these receptors are mutually exclusive from those expressing *fru-LEXA* within the same bristle [20]. This accords with the scRNA-seq data, where *fru* appeared largely restricted to the male- and female-sensing populations (Fig 7F and 7AH). Sexual dimorphism in these neurons is therefore instead likely driven by *dsx*; indeed, at least some *Ir52c*+ neurons have been shown to descend from a *dsx*+

lineage [20]. In the adult dataset, *dsx* expression was patchy among GRNs, appearing enriched in the male- and female-sensing populations (S11Y Fig). In the pupal data, *dsx* was widely detected across all GRN clusters (S11Z Fig). The discrepancy in the extent of *dsx* expression in GRNs between the datasets may reflect the differences in the dissected regions: While most male and female pheromone-sensing GRNs may be *dsx*+ regardless of position, it may be that *dsx* expression is restricted among *nvy*+ and *fkh*+ neurons to those in the regions of the foreleg more likely to contact a mate than food—i.e., the foreleg upper tarsal segments. Restriction of *dsx* expression to a subset of neurons within a GRN class would provide a mechanism through which an additional layer of between-bristle variation in activity could be achieved.

## Distinct and shared modules of gene expression in *nvy*+ and *fkh*+ GRNs

The 3 GRN populations that we have discussed—male-sensing, female-sensing, and *nvy*+—match known populations in the literature. However, we were unable to find mention of a population that resembled our fourth, which was *pros*+/*acj6*+/*fkh*+. Although none of the top 20 DEGs obtained from a comparison with the 3 other GRN populations showed specific expression in the pupal data, there were intriguing similarities with the *nvy*+ cluster: *CAH2*, *jus*, and *Glut4EF* each looked specific to or highly enriched in both the *nvy*+ and *fkh*+ GRNs (S11AA–S11AC Fig). But in the adult dataset, these specific differences were reduced or lost, suggesting that the variation we observed between GRNs in the pupal data may reflect heterochronic differences or that these particular between-GRN developmental differences are lost in adulthood (S11AD–S11AF Fig). Nonetheless, the adult dataset presented its own similarities between *fkh*+ and *nvy*+ GRNs: Expression of *Ir76b*, which is known to be widely expressed among olfactory and gustatory receptor neurons and is thought to form heteromeric complexes with more selectively expressed *Ir's*, was restricted to *fkh*+ and *nvy*+ GRNs (S11AG Fig) [20,129–131]. The restricted expression of *Ir76b* contrasts with that of another such coreceptor, *Ir25a*, which we found broadly expressed across all 4 GRNs (S11AH Fig). Although detected in fewer cells, *Ir40a*, which is known to be coexpressed with *Ir25a* in the antennal sacculus, was similarly broadly expressed [132] (S11AI Fig).

The *fkh*+ population also had something in common with the female-sensing GRNs: Across both datasets, the extracellular matrix proteoglycan gene *trol*—which we also observed in tendon cells (S6S–S6V Fig)—was enriched in *fkh*+ and female-sensing GRNs, showed patchy, low-level expression in male-sensing GRNs, and was essentially absent from *nvy*+ GRNs (S12A–S12D Fig). We recovered this pattern in *trol-GAL4 > UAS-mCD8::GFP* [133] first tarsal segments counterstained with anti-Pros: Of the Pros+ cells in a single chemosensory bristle, at least 2 were strongly *trol-GAL4*+ and at least 2 were *trol-GAL4*− (including the sheath) (S12E–S12G Fig). In some bristles, the remaining Pros+ cell was *trol-GAL4*+ and in others *trol-GAL4*−. *trol* performs several roles during the assembly of the nervous system [134,135]—why it should be limited in its expression among GRNs is unclear. Alongside these shared modules of gene expression, we identified several uniquely expressed genes in *fkh*+ GRNs, including *Tbh*, which encodes the key limiting enzyme in octopamine synthesis and therefore suggests that these neurons are octopaminergic (S11AJ and S11AK Fig) [136]. Although very sparsely detected, we also observed *Ir60a* to be limited to *fkh*+ neurons (S11AL Fig).

## Potential heterochrony in the gene expression profiles of sensory organ neurons

In attempting to identify subtype-specific genes among subclasses of neurons, we observed that genes that appeared cluster-specific in the pupal stages sometimes showed widespread expression in the adult nuclei data. For example, *DCX-EMAP*, a gene that has been implicated

in mechanotransduction in both campaniform sensilla and the chordotonal receptors of the Johnston's organ [137], switches from being exclusive to campaniform sensilla neurons in the pupal data, to being widespread among both campaniform sensilla and mechanosensory neurons in the adult data (S13C and S13G Fig). Several other genes, such as *nAChRalpha7*, *Ccn*, *CG1090*, *CG17839*, and *CG34370*, showed a similar pattern (S13A–S13N Fig). Analogously, MSNCBs and sex comb neurons, identifiable as a $vvl^-$ and $vvl^+/fru^+/rn^+$ subpopulation of mechanosensory neurons, respectively, appeared to develop ahead of the major body of mechanosensory neurons that innervate mechanosensory bristles, as evidenced by the expression patterns across datasets of genes such as *sosie*, *CG31221*, and *dpr13* (S13O–S13T Fig). In these cases, the broadening of expression between pupal and adult datasets is suggestive of heterochronic differences (i.e., a difference in rate, timing, or duration) in development between the neurons of different sensory organ classes, such that the neurons in mechanosensory bristles lag behind those in chemosensory bristles, campaniform sensilla, and the sex comb. This hypothesis is consistent with documented between-organ variation in developmental timing: Chemosensory bristles are known to be specified earlier than all but the largest mechanosensory bristles [115,116].

## Mechanotransduction neurons from different external sensory organ classes express largely shared gene repertoires

Heterochronic differences between sensory organs complicate the identification of organ-specific genes in the pupal data: Genes that appear unique at one stage may be widespread at a later time point. For that reason, we initially focused on the adult nuclei dataset to identify genes enriched in specific non-GRN neuron populations (Fig 7AI). The majority of the top mechanosensory neuron markers that were returned from a DGE analysis comparing these cells to all other neurons appeared nonspecific, being expressed in one or more additional clusters (e.g., *Calx*, *Fife*, *Dop2R*, *KrT95D*, *CG4577*, and *Ten-m*; S14A–S4M Fig). The same applied to the top campaniform sensilla markers, but in this case, despite their lack of complete specificity, many showed a relatively restricted expression profile that extended across both campaniform sensilla and chordotonal organs (e.g., *dati*, *unc79*, *CG42458*, *TyrR*, *Cngl*, *CARPB*, and *beat-VI*; S14N–S14Z Fig). This pattern is suggestive of certain molecular commonalities between campaniform sensilla and chordotonal organ neurons, commonalities that make them distinct from other mechanotransduction neurons. In further pursuit of genes specific to each of mechanosensory neurons and campaniform sensilla, we tried another approach: identifying top markers of each of these clusters in the pupal data and mapping their expression in adults to determine whether their expression remains cluster specific. Of these, the transcription factor *Ets65A*, which in the pupal data was restricted to mechanosensory neurons, sex comb neurons, MSNCBs, and campaniform sensilla neurons, remained restricted to mechanosensory neurons and campaniform sensilla neurons in the adult data (S14AA and S14AB Fig). *Ets65A* therefore represents a candidate regulator of mechanosensory identity in external sensory organ neurons.

We applied this same pupal identification and adult mapping approach in sex comb neurons and MSNCBs, populations that didn't form distinct clusters in the adult data. No individual gene showed clearly restricted expression to either, but there was strong enrichment of genes that were otherwise patchily expressed across cells. *shakB* was strongly enriched in sex comb neurons relative to other mechanosensory bristles in the pupal data and widely present across putative chordotonal and campaniform sensilla neurons in the adult data (S14AC and S14AD Fig). In MSNCBs, *CG42566*, and to a lesser extent *CG33639*, appeared enriched relative to other mechanosensory populations, with both also detected across GRNs, consistent with a

chemosensory bristle origin (S14AE–S14AH Fig). Collectively, MSNCBs and sex comb neurons showed clear enrichment of both effectors of the sex determination pathway: *fru* and *dsx* (S14AI–S14AL Fig). The specificity of this enrichment was greater in the case of *dsx*, which across both neuronal datasets was largely absent outside of the comb, GRN, and MSNCB clusters; *fru* was surprisingly widespread in the adult data, including expression in the putative chordotonal clusters and campaniform sensilla. The expression of these 2 transcription factors provides a clear regulatory mechanism through which the transcriptomic profiles and activity of these derived mechanosensory populations could readily diverge from other mechanosensory bristles—and do so in a sex-specific manner. But ultimately, determining whether any of the transcriptomic differences between the mechanotransduction neurons that innervate each of mechanosensory bristles, chemosensory bristles, the sex comb, and campaniform sensilla translate into functional differences in operation or sensitivity requires further work. It's clear from these data that some differences between mechanotransduction neurons innervating different organ classes are present, but they appear minor and on the whole less clear cut than between GRN populations.

## Putative chordotonal organ neuron subtypes express specific gene repertoires

In the FCA adult data, we recovered 3 putative chordotonal organ neuron clusters that were absent from our first tarsal segment dataset because these organs fall outside the dissected area. We refer to the largest as "putative chordotonal" and the 2 smaller clusters as *erm*[+] and *bab1*[+] putative chordotonal based on a transcription factor they were each specifically enriched for. Many of the genes we identified as specifically enriched in the putative chordotonal organs in our DGE analysis were absent from the pupal dataset, consistent both with their organ specificity and the absence of chordotonal organs from our dataset. Along with the genes shared among campaniform sensilla neurons and chordotonal organ neurons discussed in the previous section (S14N–S14Z Fig), we identified genes specific to or highly enriched in all chordotonal populations (S15A–S15K Fig), as well as those specific to subpopulations (S15–S15V Fig). The transcriptomic distinctiveness we detect between putative chordotonal neuron clusters aligns with previous work that has identified multiple, functionally distinct neurons in a single chordotonal organ (e.g., "Type A" and "Type B" neurons;, [4,5]). The next step will be to match the distinct clusters we recover to these different chordotonal neuron classes. In turn, that would raise secondary questions, such as whether each neuron class is present in each of the several chordotonal organs that are housed within the leg.

## The support cells within a mechanosensory organ each express a distinct gene repertoire

Campaniform sensilla, chemosensory bristles, and mechanosensory bristles each consist of 4 distinct cell types generated through asymmetric divisions of a sensory organ precursor (SOP). The first division gives rise to a progenitor of the socket (tormogen) and shaft (trichogen) cells, while the other to a progenitor of the sheath (thecogen) and neuron(s) (reviewed in [138]) (Fig 8A). Few marker genes are known for the nonneuronal SOP descendants and, to the best of our knowledge, none that definitively separate the same cell type between different organs (e.g., mechanosensory versus chemosensory sockets) [26]. Those markers that are known include the following: *Su(H)* and *Sox15*, which specifically accumulate in socket cell nuclei [139,140]; *sv* (*Pax2*), which although initially expressed in all bristle cells during the mitotic phase of development is eventually restricted to the shaft and sheath [141]; *nompA*, which is

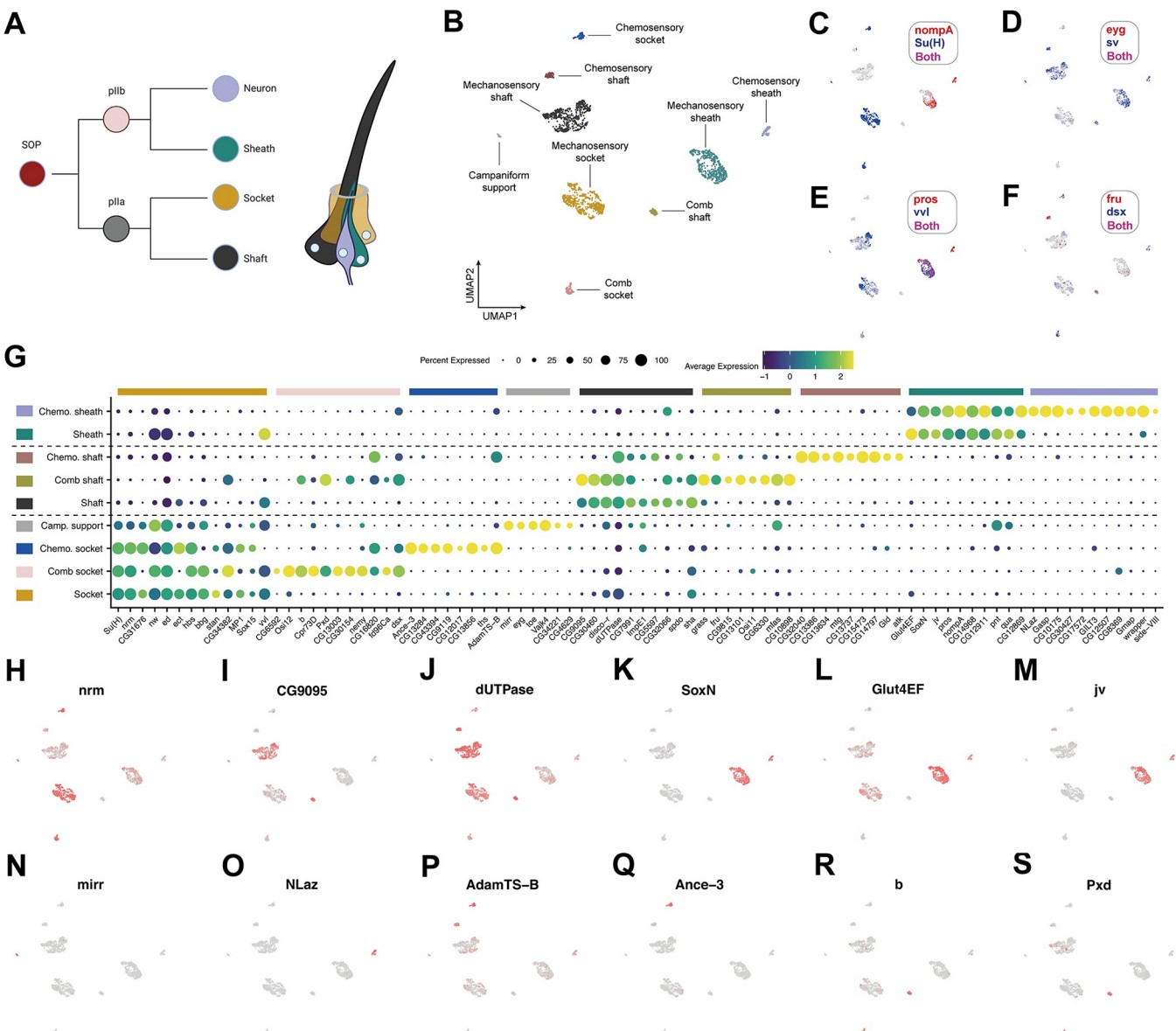

**Fig 8. Distinct and shared modules of gene expression between sensory organ support cells.** (A) The 4 constituent cell types of external sensory organs, such as the mechanosensory bristle in this schematic, originate through asymmetric divisions of a sensory organ precursor (SOP) cell (reviewed in [138]). The SOP divides to produce a pIIa and pIIb daughter cell. pIIa further divides to generate a socket and shaft cell. In the notum, where it's been studied, pIIb divides into a pIIIb cell and glial cell, the latter of which enters apoptosis soon after birth [188]. pIIIb further divides to produce the sheath and neuron. To the best of our knowledge, whether the pIIb glial division occurs in the leg remains untested. (B) Annotated UMAP plot of the bristle cells from the integrated 24 h AFP and 30 h APF first tarsal segment dataset. The campaniform support cluster included *Su(H)*+ cells, which suggests that it corresponds to socket cells, but it's possible that it includes a mix of campaniform sensilla accessory cell types. (C-F) The UMAP shown in (B) overlaid with the expression of a series of marker genes, either previously published or demonstrated in this study, for different sensilla classes or accessory cell types. (G) A dot plot summarizing the expression patterns of a selection of genes identified as being differentially expressed in each of the clusters given in the UMAP shown in (B). Dotted lines separate the 3 major classes of sensory support cell. Of the socket markers, *CG31676* is known to be expressed in a subset of olfactory projection neurons [133]; *nw* is a C-type lectin-like gene; *stan*, a cadherin that controls planar cell polarity [189]; and *nrm*, *ed*, and *hbs* are cell adhesion molecule genes [143–147]. Of the shaft markers, *CG9095* encodes a C-type lectin-like gene; *disco-r* encodes a transcription factor; *dUTPase* encodes a nucleoside triphosphate; *spdo* encodes a transmembrane domain containing protein that regulates Notch signaling during asymmetric cell division [190–192]; and *sha* encodes a protein involved in the formation of bristle hairs [148]. Aside from *pros* and *nompA*, the top markers of the sheaths include the following: the transcription factors *Glut4EF*, *pnt*, and *SoxN*; *jv*, which encodes a protein involved in actin organization during bristle growth [149]; *qua*, which encodes an F-actin cross-linking protein [150]; and the midline glia marker *wrapper*, which encodes a protein involved in axon ensheathment [99–101]. (H-S) The UMAP shown in (B) overlaid with the expression of genes identified in this study as markers of sensory organ support cell subtypes. Data and code for generating the figure are available at https://www.osf.io/ba8tf.

specifically expressed in sheath cells where it is required to connect dendrites to the shaft [142]; and *pros*, which is expressed in sheath cells (Fig 6K–6P; [113]).

Based on these markers, we identified major shaft, socket, and sheath clusters in our data (Fig 8B–8F). The large size of each of these relative to other clusters in the support cell dataset indicates that they belong to the dominant sensory organ in the first tarsal segment: mechano-sensory bristles. Further evidence for a mechanosensory origin comes from the observation that each of these clusters was *vvl*[+], which we previously observed to be expressed in all mechanosensory, but no chemosensory, bristle cells (Fig 6Q–6S). Among the DEGs we identi-fied for each of the mechanosensory shaft, socket, and sheath clusters were many that reflected the biology of these sensory support cells (Fig 8G–8M): the cell adhesion molecule genes *nrm*, *ed*, and *hbs* [143–147] in socket cells; *sha*, which encodes a protein involved in the formation of bristle hairs [148] in shaft cells; and, in the sheath, a duo of genes, *qua* and *jv*, involved in the organization of actin during bristle growth [149,150]. Another nod to the biology of the sheath came in its enrichment for *wrapper*, which encodes a protein known to be involved in axon ensheathment [101]. Classically used as a marker of midline glia [99–101], the expression of *wrapper* in sheaths reinforces their glia-like properties, despite not expressing *repo* or *gcm*. Moreover, it points to general, shared elements in the mechanisms of ensheathment of neuro-nal processes between these cell types. The enriched expression in sheaths of a trio of transcrip-tion factors—*Glut4EF*, *pnt*, and *SoxN*, of which the latter was completely restricted to sheaths in our dataset—provide a potential route to regulatory divergence from other support cells and glia.

## Homologous support cells show transcriptomic divergence between sensory organ classes

We observed that homologous support cells in different sensory organ classes express both shared and distinct gene repertoires. The top markers for each of the mechanosensory socket, shaft, and sheath clusters were enriched in a set of smaller clusters: 3 clusters showed socket-like profiles, 2 shaft-like profiles, and 1 a sheath-like profile (Fig 8G). These minor clusters therefore appear to be sensory organ cell subtypes from sensilla classes that are less abundant in the first tarsal segment than are mechanosensory bristles. Our visualization of *eyg-GAL4 > UAS-GFP* revealed *eyg-GAL4* expression in all 4 cells in a campaniform sensillum (Fig 6AA), so the presence of *eyg* and *toe* in the smallest cluster suggest that these cells correspond to a campaniform sensilla population. This cluster uniquely expressed the transcription factor *mirr* (Fig 8N) and the CPLCP cuticle protein family gene *Vajk4*, and was *Su(H)*[+], suggesting a socket cell identity, but no further *eyg*[+]/*toe*[+] clusters were present. Several explanations for why are plausible: (a) this cluster contains a mix of all campaniform sensilla support cells; (b) only the sockets are transcriptomically distinct enough to cluster separately; and (c) only the sockets were recovered in sufficient numbers to cluster separately. That a small, coclustering group of *eyg*[+] sheath cells were present in the mechanosensory sheath population, rather than forming their own distinct cluster, provides some support for (b) and (c) (Fig 8D). Ultimately, the presence of *eyg* and *toe* across the constituent cells of campaniform sensilla suggest that these genes may be master regulators of campaniform sensillum identity.

The minor sheath population was enriched for *CG42566*, a gene we previously found to be specific to MSNCBs and GRNs among the neurons (S14AE and S14AF Fig), suggesting that this cluster represents descendant cells of the chemosensory pIIb lineage and, therefore, the chemosensory sheath cells specifically. As well as several poorly characterized genes, this clus-ter showed enriched or unique expression of the lipid-binding protein encoding gene *NLaz* (Fig 8O), the chitin-binding protein encoding gene *Gasp*, and *Side-VIII*. As we observed with

neurons, at least some of the differences between mechanosensory and chemosensory sheaths appear to be heterochronic, with chemosensory sheaths developing ahead of their mechanosensory homologs: Several genes, including *nompA* and *wrapper*, were widely detected among chemosensory sheath cells, but in mechanosensory sheaths showed localized expression in a region enriched for cells from the 30 h dataset (S16A–S16J Fig).

Based on the expression of *Su(H)*, *Sox15*, and *sv*, the remaining clusters have apparent socket and shaft identities. Given the representation of different sensilla classes in the first tarsal segment, the most likely classification for the $vvl^-$ clusters are chemosensory sockets and shafts. Except for the extracellular protease gene *AdamTS-B*, which was heavily enriched in both clusters (Fig 8P), there was no clear transcriptomic link between them. Among the top markers for each, the $sv^+$ putative shaft cluster showed strong enrichment for *mtg*, which encodes a chitin binding domain-containing protein that's required to drive postsynaptic assembly [151], while the $Su(H)^+$/$Sox15^+$ putative socket cluster showed unique expression of *CG43394* (S17 Fig) and *Ance-3* (Fig 8Q), which encodes a predicted membrane component orthologous to human ACE2, the receptor for SARS-CoV-2 [152].

### Sex comb cells are enriched for the expression of melanogenic pathway genes

Unlike the chemosensory support cells, the remaining minor putative shaft and socket clusters shared many of their top markers with one another. These genes included the following: *b* (Fig 8R) and *Pxd* (Fig 8S), components of the melanogenic pathway [153,154]; *dsx*, the effector of the sex determination pathway (reviewed in [64]); and *fd96Ca*, a gene known to contribute to sex comb formation [155] and which we found to be one of the top markers of sex comb neurons in the pupal data. These 2 clusters therefore likely correspond to sex comb shafts and sockets. Whether the genes we find enriched in sex comb sockets and shafts (Fig 8G) are unique to these cells is hard to determine from these data alone due to heterochronic differences between mechanosensory bristles and the sex comb, the latter of which develops earlier. This difference is reflected in the expression of *vvl*, which was substantially reduced in the putative sex comb shaft cluster relative to both the sex comb socket and the mechanosensory populations (Fig 8E and 8G). Previous work has shown that while Vvl is present in all 4 SOP-descendent cells in external sense organs on the head and notum at 24 h APF, by approximately 42 h, it's restricted to the socket [156]. Despite the heterochronic differences that likely exist, the enrichment of melanogenic pathway genes is consistent with the sex comb's heavily melanized appearance. Further work is required to determine whether the changes to mechanosensory bristle cells that generate the modified sex comb morphology operate primarily through quantitative, qualitative, or heterochronic changes in gene expression.

### Discussion

Sensory perception begins with contact between a stimulus and a specialized sensory organ. Within and between animal species, these organs are highly variable in their form, cellular composition, and molecular characteristics. To understand the genetic underpinnings of this diversity, both in terms of the developmental networks that specify organ development and the molecular profiles that define mature organs, we characterized the transcriptomes of cells in a region of the male *Drosophila* foreleg that carries a structurally and functionally diverse selection of sensory organs.

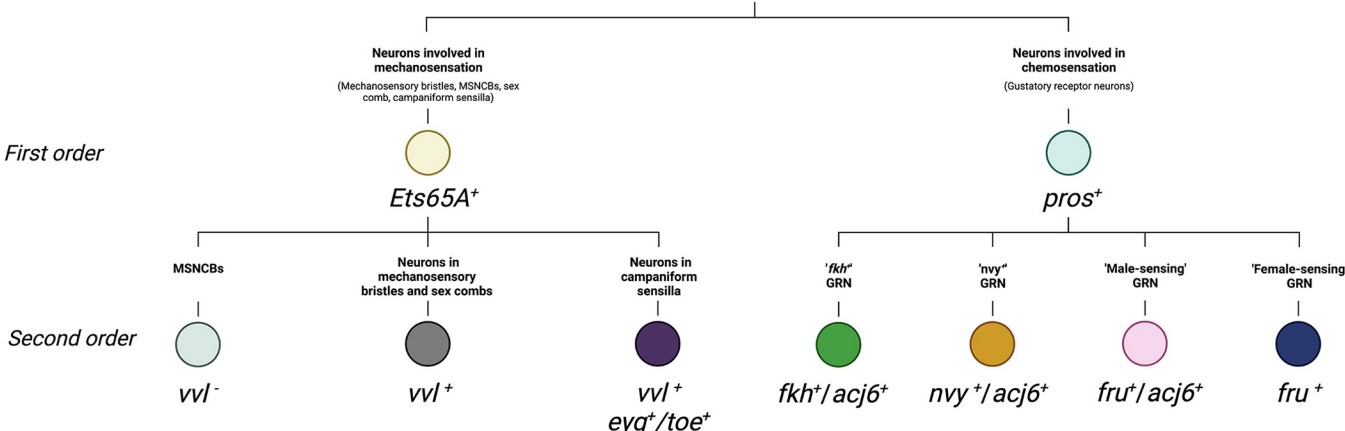

**Fig 9. First and second order differences in transcription factor expression between sensory neuron classes.** GRN, gustatory receptor neuron; MSNCB, mechanosensory neuron in chemosensory bristle.

## Hierarchical and combinatorial transcription factor codes for leg sensory neurons

The discrete identities adopted by cell types depend on cell type–specific expression of transcription factors, which function as regulators of downstream networks of gene expression. We identified a combinatorial transcription factor code unique to each sensory neuron population present in the first tarsal segment (Fig 9). The "first order" differences in transcription factor expression defined neurons involved in mechanosensation (i.e., those innervating mechanosensory bristles, the sex comb, and campaniform sensilla, along with the MSNCB), which expressed *Ets65A*, and neurons involved in chemosensation, which expressed *pros*. The "second order" differences defined subtypes within each of these major classes: The expression of *eyg* and *toe* separated campaniform sensilla from other mechanotransduction neurons; *acj6*, *fru*, *nvy*, and *fkh* delineated GRN subtypes; and *vvl* separated the mechanosensory neurons of mechanotransduction organs from those that innervate chemosensory bristles. Of these transcription factors, *acj6*, and to a lesser extent *fkh*, have repeatedly cropped up in studies of *Drosophila* neurons, including in subpopulations within the visual, auditory, and olfactory systems [30,47,157–160]. The expression of these transcription factors in a subset of functionally varied neurons points to shared regulatory architecture between them and, by extension, perhaps to common downstream networks of gene expression.

## Orthogonal regulatory input from the sex differentiation pathway?

The effectors of the sex determination pathway, *fru* and *dsx*, showed restricted expression across neuron populations. Two GRNs expressed *fru*, as did a subpopulation of mechanosensory neurons innervating the sex comb. *dsx*, on the other hand, was expressed in all GRN populations, sex comb neurons, and in the subpopulation of mechanosensory neurons that innervate chemosensory bristles (MSNCBs). However, the prevalence of *fru* and *dsx* expression differed among neuron classes. *fru* was widely detected in female- and male-sensing GRNs and the sex comb neurons but was only present in a few *fkh*+ and *nvy*+ GRNs. In the pupal first tarsal segment data, *dsx* was detected across all GRNs, although more patchily in *fkh*+ neurons than in the other subtypes. In the adult GRNs, *dsx* was more widely detected among female- and male-sensing GRN clusters with a very small number of *dsx*+ cells present in the remaining GRN populations. The patchiness of expression may reflect gene dropout,

with differences in patchiness between populations stemming from quantitative differences in expression. Another explanation, however, is that in some neuron classes, *fru* and *dsx* expression is limited to subsets of cells. In this case, sexual identity would serve as a regulatory input orthogonal to the neuron's core genetic identity and, by extension, provide a regulatory route to sexually dimorphic gene expression in a subset of neurons within a shared class. The orthogonal nature of this regulatory input is most likely to manifest itself in the $dsx^+$ or $fru^+$ subsets of $fkh^+$ and $nvy^+$ GRNs, where *dsx* and *fru* expression appears to be relatively rare. In contrast, in the male- and female-pheromone sensing GRNs or the sex comb, *fru* expression may form a core part of that cell population's identity.

## Reconciling the functional diversity of GRNs with their limited transcriptomic diversity

Considerable diversity is known to exist among leg chemosensory taste bristles in the sensitivities and responsiveness they show to a wide range of tastants, as well as in the identities of the receptors they express [3,19,20,125]. But in all of the datasets we analyzed here—two from a highly localized region of the developing foreleg and one from the entire length of each of the 3 pairs of adult legs—we resolved only the same 4 GRN populations. How, then, can we reconcile the functional diversity known to exist between leg taste bristles with the limited number of transcriptomically distinct GRN classes that we identified? There are several possible solutions. First, a member of each GRN class might not be present in every bristle on the leg, but rather members of a class may be restricted to subsets of bristles. As each chemosensory bristle on the tarsal segments is thought to be innervated by 4 GRNs [16], this would require that some bristles are innervated by multiple GRNs of the same class. Second, it may be that there are additional functionally and transcriptomically distinct GRN classes in the leg, but that they are rare in comparison to the 4 classes we resolved. A very small number of taste bristles at the tips of the leg are known to not be innervated by a $ppk25^+$ female-sensing GRN [125]. This same region also houses some of the least numerous taste bristle classes—classes that are more likely to contact food than potential mates—that have been identified on the basis of morphology and response to tastants [19]. Bristles in this region of the leg would naturally not have made it into our first tarsal segment datasets and their rarity in the wider leg may have limited their recovery in the adult dataset.

There is a third solution to the disconnect between the known functional diversity and transcriptomic similarity that we recover among GRN populations. It might be that the leg taste bristles are largely—or in every case—innervated by a minimal number of 4 functionally distinct GRN classes, each defined by a unique combination of transcription factors. But on top of this core transcriptomic program may be layered an additional level of regulation, one that we were unable to resolve here, which enables the same GRN class in different bristles to express a different receptor or membrane channel repertoire. There is evidence for this in the datasets we looked at here. *Ir52c* and *Ir52d*, 2 genes expressed in our $nvy^+$ population, have been shown by both reverse transcription PCR (RT-PCR) and GAL4s to be exclusively expressed in the forelegs, where they are coexpressed [20]. In a dataset composed of all 3 pairs of legs, like the adult nuclei dataset we analyzed, $Ir52c^+/Ir52d^+$ cells should therefore comprise a very small number of the total recovered cells, their numbers having been diluted by GRNs from bristles on the other legs. This is indeed what we see, but rather than forming their own cluster, we find the $Ir52c^+/Ir52d^+$ cells alongside a greater number that are negative for these 2 genes in a cluster that also includes cells expressing *Ir52a*, which is known to be expressed in bristles on all legs [20]. Expression patterns such as this, where cells expressing a different set of receptors cocluster, along with the similar representation of cells from each GRN class in

our datasets suggest that a "minimal GRN class" model seems likely to underlie most, if not all, taste bristles on the leg.

The nature of the extra regulatory layer that allows receptors to be swapped in and out of a core GRN class remains unclear. The positions of the bristles in which some receptors show restricted expression appear stereotyped [3,19], which suggests that additional layers of regulation may involve positional information. It also suggests that any role for stochasticity in receptor expression, as occurs in mammalian olfactory neurons [161], might be limited. Additionally, restricting the expression of the effectors of sex determination to subsets of bristles would allow for regulatory divergence in neurons from a common GRN class between bristles.

Whatever the differences between the neurons of a single GRN class innervating different bristles may be, our data suggest that the identity of a GRN is more than just the receptors it expresses and that there are deeper transcriptomic features that define a common GRN class. The defining features of a GRN class may relate to the neurotransmitters through which they relay the detection of stimuli or, perhaps, the complement of IR coreceptors they express. On this point, some IRs, including IR25a and IR76b, are known to act as cofactors, forming heteromeric complexes with more selectively expressed IRs [20,129–131]. We found these coreceptors to vary in their specificity, *Ir25a* being broadly expressed across all 4 GRNs and *Ir76b* restricted to *fkh*+ and *nvy*+ GRNs. We could imagine a modular configuration where coreceptors could set the broad functional range that defines a GRN class—i.e., those classes we recover in our single-cell data—while additional gene products, differentially expressed between neurons of a given GRN class in different bristles, refine GRN function to generate functional diversity between bristles.

Three of the 4 GRN classes we identify match the descriptions of functional populations that have been reported in the literature, each of which is involved in the detection of conspecifics. Two of these are the *fru*+/*ppk23*+ "female-pheromone-sensing" and "male-pheromone-sensing" neurons (also known as "F cells" and "M cells"), a pair of neurons that respond selectively to the pheromones of the sex they're named after [22]. Of the genes we identify as specifically expressed in these cells, *ppk10* and *ppk15* stand out as candidate male pheromone detectors, given their restriction to male-sensing neurons and the known roles of other *ppk* genes in pheromone sensing (e.g., [21,123–126]). The other identifiable GRN population that we resolved was the *nvy*+, *Ir52a,c,d*-expressing neurons [20]. These neurons are activated by exposure to conspecific females, but not males, and promote courtship. In this sense, naming *fru*+/*ppk23*+/*ppk25*+ GRNs as "female pheromone-sensing neurons"—a name used by others [22] and by us throughout this paper—fails to recognize that these neurons can sit alongside a different GRN with apparently the same broad function in the same bristle. The existence of 2 distinct classes of female-sensing neurons raises the question of why two should be required. That we find *ppk* genes to be limited to one of the 2 female-sensing GRNs points to differences in either what the neurons respond to or how they respond. A large number of putative pheromones exist in female *Drosophila*, both in *D. melanogaster* and in other species [162,163]. This diversity might necessitate multiple detector neurons in order to accurately discriminate between sex and species and perhaps to even allow males to perform finer-grain assessments of the condition or mating status of potential partners. Future work will be required to determine whether responsiveness to these different compounds is divided between the 2 GRN classes, whether there is a logic to that division based on the identity of those compounds, and whether these GRNs project into different neural circuits. Finally, that leaves the 1 GRN population—the *fkh*+ population—that we couldn't identify. Whether this class also contributes to conspecific detection and to what extent its function may vary between bristles remains unclear.

## The not-so-specific expression of glia markers

Three glia cell types have been described in the *Drosophila* peripheral nervous system: (1) wrapping glia, which ensheath individual or bundled axons to support the rapid conductance of action potentials; (2) perineural glia, which form the outer cell layer of the nervous system, positioned below a dense network of extracellular matrix called the neural lamella, and which are thought to be responsible for nutrient uptake via the contacts they make with hemolymph; and (3) subperineural glia, large cells that form a thin layer beneath the perineural glia and which establish septate junctions with one another to provide an important structural component of the blood–brain barrier (reviewed in [104,164]). All 3 of these glia are known to be present in the leg [89], and a number of marker genes are known for each [95–97,165,166]. But the glia populations we resolved did not fall neatly along those marker gene lines. *moody* and, to a lesser extent, *Gli*, which are both used as specific markers of subperineural glia, were widely detected across our *repo*⁺ cells, as were the wrapping glia markers *nrv2* and *Ntan1* [89,96]. *Jupiter*, which is thought to specifically mark perineural glia [96], was even more widely detected, being expressed in all nonsensory cells. The noncanonical expression patterns we observe in our glia populations with respect to known marker genes raise difficult questions. Have the GAL4s and protein traps we've relied on given us a misleading impression of glia type-specific patterns of gene expression and protein localization? Perhaps—it's clear that for at least some of these markers antibodies label a wider set of cells than do drivers, as in the case of the *apt-GAL4 GMR49G07* [89]. But the surprisingly broad expression of known marker genes is also consistent with an emerging pattern in single-cell studies, where the expression of many genes is considerably wider than expected from reporter and antibody staining data (e.g., [35]). This phenomenon illustrates the importance of cell type–specific posttranscriptional regulation in enforcing the cell type specificity of gene expression, as has been shown for surprisingly widely expressed genes such as the neuronal marker *elav* [114]. After all, mRNA levels are not the final output of gene expression [167].

## A novel cell type and the construction of the neural lamella

We identified a population that didn't seem to match the description of any reported in the literature. These cells, enriched for the expression of the transcription factors *Sox100B* and *Lim1*, exhibited several glia-like properties. For one, they expressed the midline glia marker *wrapper* [99–101]. More directly, our stainings showed a glia-like association between these cells and both the axon trunks that project from the leg sensory organs to the VNC and the VNC neuropils. Enriched expression of the *beaten path* genes, *beat-IIa* and *beat-IIb*, was further suggestive of a cell that's in direct communication with others in the developing nervous system [107]. But from a glia perspective, these cells pose a problem: They're negative for the canonical glia marker *repo*, a transcription factor expressed in all glia except for those of the midline, and they appear to originate within the leg disc itself, rather than migrating in along with CNS-derived glia. We have 3 clues for what these cells might be. First, a homolog of *Sox100B*, *Sox10*, is involved in the specification of glia cells that play similar roles to *Drosophila* wrapping glia in the vertebrate nervous system (oligodendrocytes and Schwann cells) [168–172]. Indeed, *Drosophila* Sox100B can rescue vertebrate Schwann cell development in the absence of Sox10 [173]. The failure of *Sox100B*⁺ cells to express the wrapping glia marker *Oaz* would argue against these cells corresponding to wrapping glia, as would their failure to express *repo*. It may therefore be the case that a conserved developmental program drives wrapping-type morphologies in both these cells and the vertebrate glia subtypes. Which leads us to the second clue: While the expression of membrane-bound GFP revealed narrow cell morphologies when driven by *repo-GAL4*, under the control of *Lim1-GAL4*, the staining appeared to encircle a

larger area, extending more laterally. The *Lim1*[+]/*Sox100B*[+] cells therefore appear to comprise the outer most layer of the nervous tissue. Which leads us to the third clue: These cells were enriched for *vkg*, a subunit of Collagen IV that's known to label the neural lamella [96,104]. Collectively, therefore, these data suggest that the *Sox100B*[+] cells are a novel, axon-associated cell type that's required for the construction of the neural lamella.

Our description of *Sox100B*[+] cells challenges the idea [174] that the source of the neural lamella is the perineural glia (see also [96], who fail to find an effect of RNAi knockdown of *vkg* on the neural lamella using glia drivers). A more recent alternative hypothesis for the formation of the neural lamella is that its major components—including Collagen IV—are deposited by migrating hemocytes during embryogenesis [96]. This idea is based on the finding that the failure of hemocytes to migrate adjacent to the developing ventral nerve cord is associated with the failure of Collagen IV and another extracellular matrix component, Peroxidasin, to be deposited around it [106]. Two things are interesting to note here. The first is that with the possible exception of *NimC3*, *Sox100B*[+] cells bear no clear transcriptomic similarity to hemocytes. The second is that hemocytes and *Sox100B*[+] cells show partially overlapping expression of many extracellular matrix component genes. It's therefore possible that both hemocytes and *Sox100B*[+] cells are collectively required for the construction of the neural lamella during pupal development, each contributing a subset of extracellular matrix components. Resolving the specific role played by *Sox100B*[+] cells in this process, as well as how labor may be divided between additional cell types, represents a key focus for future work.

## Materials and methods

### Fly strains and husbandry

Flies used in all experiments were raised on a standard cornmeal medium and housed in an incubator at 25˚C on a 12:12 cycle. Lines used in this study are detailed in Table 1.

### Isolation of first tarsal segments for single-cell sequencing

The following procedure was used to collect known-age first tarsal segments for single-cell sequencing. White P1 prepupae (0 to 1 h APF) from the DGRP line *RAL-517* [175] were collected and sexed based on the presence of testes. Males were transferred to a folded kimwipe, wet with 500 µl of water, and held inside of a petri dish in an incubator maintained at 25˚C on a 12:12 cycle. Pupae were removed from puparia using forceps and placed on top of a water-soaked kimwipe, 1 h before the desired age was reached (e.g., 23 h after collection for the 24 h sample). When the desired age was reached, pupae were placed ventral side up on tape. The base of the abdomen was pierced to release some of the fluid pressure and the foreleg removed at the tibia/tarsal joint. The dissected leg was then placed on tape, covered in a drop of 1X Dulbecco's PBS (DPBS, Sigma, D8537), and a Micro Knife (Fine Science Tools, 10318–14) was used to sever at approximately the midpoint of the second tarsal segment. The first tarsal segment was then eased out of the pupal cuticle and transferred to a glass well on ice containing 100 µl of 1X DPBS using a BSA-coated 10-µl tip.

### Single-cell suspension preparation

We separately prepared 2 single-cell suspensions, each composed of 67 first tarsal segments collected following the procedures described above: one from 24 h APF male pupae, the other from 30 h APF male pupae. For a given time point, once 67 first tarsal segments had been collected, the DPBS was removed from the well and replaced with 100 µl of dissociation buffer, which consisted of 10X TrypLE (Thermo Fisher Scientific, A12177-01) with a final

**Table 1. Reagents and resources used in this study.**

| Reagent type or resource | Designation | Source or reference | Identifiers | Additional information |
|---|---|---|---|---|
| Genetic reagent (*D. melanogaster*) | *RAL-517* | [175] | RRID:BDSC_25197 | |
| Genetic reagent (*D. melanogaster*) | *10XUAS-IVS-mCD8::GFP* | Bloomington Drosophila Stock Center | RRID:BDSC_32188 | |
| Genetic reagent (*D. melanogaster*) | *UAS-mCherry.nls* | Bloomington Drosophila Stock Center | RRID:BDSC_38424 | |
| Genetic reagent (*D. melanogaster*) | *UAS-GFP.nls* | | | |
| Genetic reagent (*D. melanogaster*) | *TfAP-2-GFP.FPTB* | Bloomington Drosophila Stock Center | RRID:BDSC_83382 | |
| Genetic reagent (*D. melanogaster*) | *nub-GAL4* | Bloomington Drosophila Stock Center | RRID:BDSC_86108 | |
| Genetic reagent (*D. melanogaster*) | *vkg-GFP (CC00791)* | [182] | | |
| Genetic reagent (*D. melanogaster*) | *Lim1-GAL4, UAS-GFP::mCD8* | Gift from Makoto Sato [102] | | |
| Genetic reagent (*D. melanogaster*) | *repo-GAL4* | Bloomington Drosophila Stock Center | RRID:BDSC_7415 | |
| Genetic reagent (*D. melanogaster*) | *fru-GAL4* | Bloomington Drosophila Stock Center | RRID:BDSC_66696 | |
| Genetic reagent (*D. melanogaster*) | *eyg-GAL4; UAS-GFP.S65T* | Bloomington Drosophila Stock Center | RRID:BDSC_58458 | |
| Genetic reagent (*D. melanogaster*) | *trol-Gal4/FM7 GMR-YFP; UAS-mCD8-GFP(II)* | Gift from Liqun Luo [133] | RRID:DGGR_113584 | Identifier is for the original GAL4 line |
| Genetic reagent (*D. melanogaster*) | *elav-GAL4* | Bloomington Drosophila Stock Center | RRID:BDSC_458 | |
| Genetic reagent (*D. melanogaster*) | *1151-GAL4* | Gift from Cedric Soler [84] | | |
| Genetic reagent (*D. melanogaster*) | *CG43394-GAL4* | Bloomington Drosophila Stock Center | RRID:BDSC_77109 | |
| Antibody | Mouse anti-Acj6 | Developmental Studies Hybridoma Bank | RRID:AB_528067 | 1:10 in 2% goat serum with TNT |
| Antibody | Mouse anti-Futsch | Developmental Studies Hybridoma Bank | RRID: AB_528403 | 1:100 in 2% goat serum with TNT |
| Antibody | Mouse anti-Prospero | Developmental Studies Hybridoma Bank | RRID:AB_528440 | 1:50 in 2% goat serum with TNT |
| Antibody | Rabbit anti-Mef2 | Developmental Studies Hybridoma Bank | RRID:AB_2892602 | 1:1,000 in 2% goat serum with TNT |
| Antibody | Rabbit anti-Sox100B | Gift from Steve Russell | | 1:1,000 in 2% goat serum with TNT |
| Antibody | Rat anti-Vvl | Gift from Sarah Certel | | 1:200 in 2% goat serum with TNT |
| Antibody | Guinea Pig anti-Fkh | Gift from Vidya Chandrasekaran | | 1:2,000 in 2% goat serum with TNT |
| Antibody | Rabbit anti-Nvy | Gift from Richard Mann [183] | | 1:300 in 2% goat serum with TNT |
| Antibody | Rabbit anti-Oaz | Gift from Marc Freeman and Megan Corty [95] | | 1:500 in 2% goat serum with TNT |
| Antibody | Anti-mouse AlexaFluor488 | Thermo Fisher Scientific | Thermo Fisher Scientific Cat# A-11001, RRID:AB_2534069 | 1:400 in 2% goat serum with TNT |
| Antibody | Anti-mouse AlexaFluor594 | Thermo Fisher Scientific | Thermo Fisher Scientific Cat# A-11005, RRID:AB_2534073 | 1:400 in 2% goat serum with TNT |
| Antibody | Anti-rat AlexaFluor568 | Thermo Fisher Scientific | Thermo Fisher Scientific Cat# A-11077, RRID:AB_2534121 | 1:400 in 2% goat serum with TNT |

(*Continued*)

**Table 1.** (Continued)

| Reagent type or resource | Designation | Source or reference | Identifiers | Additional information |
|---|---|---|---|---|
| Antibody | Anti-rabbit AlexaFluor488 | Thermo Fisher Scientific | Thermo Fisher Scientific Cat# A-11008, RRID:AB_143165 | 1:400 in 2% goat serum with TNT |
| Antibody | Anti-guinea pig AlexaFluor594 | Thermo Fisher Scientific | Thermo Fisher Scientific Cat# A-11076, RRID:AB_2534120 | 1:400 in 2% goat serum with TNT |
| Antibody | Anti-rat FITC | Santa Cruz Biotechnology | Santa Cruz Biotechnology Cat# sc-2011, RRID:AB_631753 | 1:400 in 2% goat serum with TNT |
| Antibody | Anti-rabbit TexasRed | Thermo Fisher Scientific | RRID: AB_2556776 | 1:400 in 2% goat serum with TNT |
| Software, algorithm | Fiji | National Institutes of Health | RRID:SCR_002285 | |
| Software, algorithm | Illustrator | Adobe | RRID:SCR_010279 | |
| Software, algorithm | CellRanger (v4.0.0) | 10x Genomics | RRID:SCR_017344 | https://support.10xgenomics.com/single-cell-gene-expression/software/pipelines/latest/what-is-cell-ranger |
| Software, algorithm | Seurat (v4.0.5) | [176–179] | RRID:SCR_016341 | https://satijalab.org/seurat/index.html |
| Software, algorithm | DoubletFinder (v2.0.3) | [181] | RRID:SCR_018771 | https://github.com/chris-mcginnis-ucsf/DoubletFinder |
| Software, algorithm | R | [184] | | |
| Software, algorithm | RStudio | [185] | | |
| Software, algorithm | SCopeLoomR | | | https://github.com/aertslab/SCopeLoomR/ |
| Reagent | Dulbecco's PBS, DPBS | Sigma | | |
| Reagent | TrypLE 10× | Thermo Fisher Scientific | D8537 | |
| Reagent | Collagenase | Sigma | A12177-01 | |
| Reagent | Chromium Next GEM Single Cell 3′ kit v3.1 | 10x Genomics | C0130 | |

concentration of 2 mg/mL of collagenase (Sigma, C0130). The well was sealed and submerged in a metal bead bath in an incubator at 37°C for 35 min. The dissociation buffer was then removed from the well and replaced with 50 μl of room temperature DPBS before the tissues were subjected to mechanical dissociation. For this, the solution was pipetted up and down 20 times using a 200-μl, widebore, low bind, freshly BSA coated tip (Thermo Fisher Scientific, 2069G) on a 50-μl pipette set to 40 μl. The tip was fully submerged to avoid bringing air into the suspension. The solution was then pipetted up and down a further 20× using a flame-rounded, BSA-coated 200-μl tip, again on a 50-μl pipette set to 40 μl. Next, the solution was pipetted up and down slowly 3 times using the same flame-rounded tip before 40 μl was taken up and transferred to a 2-mL, low-bind, wide-bottomed tube on ice. Exactly 20 μl of DPBS was then slowly dripped around the edges of the well using a different tip to flush the cells into the center. This was then pipetted up and down 3 times using the original flame-rounded 200-μl tip and added to the tube. Cell concentration and viability was assayed using an acridine orange/propidium iodide stain and measured using a LUNA-FL Fluorescent Cell Counter averaged across $2 \times 5$ μl aliquots (24 h APF sample: 1,990 cells/μl, 98% viability; 30 h APF sample: 967 cells/μl, 98% viability). Through this approach, we were able to avoid filtration or centrifugation steps, both of which we observed to result in high cell loss in trial runs. Although we didn't perform a controlled test of it, we also observed that the viability of our cell suspensions was considerably higher when using DPBS compared to PBS.

## Library preparation and sequencing

Barcoded 3′ single-cell libraries were prepared from single-cell suspensions using the Chromium Next GEM Single Cell 3′ kit v3.1 (10X Genomics, Pleasanton, CA) for sequencing according to the recommendations of the manufacturer. The cDNA and library fragment size distribution were verified on a Bioanalyzer 2100 (Agilent, Santa Clara, CA). The libraries were quantified by fluorometry on a Qubit instrument (LifeTechnologies, Carlsbad, CA) and by qPCR with a Kapa Library Quant kit (Kapa Biosystems-Roche) prior to sequencing. The libraries were sequenced on a NovaSeq 6000 sequencer (Illumina, San Diego, CA) with paired-end 150-bp reads. The sequencing generated approximately 50,000 reads per cell and 500 million reads per library.

## scRNA-seq data processing

The alignment, barcode assignment, and UMI counting of the two 10× leg samples were performed using the "count" function in CellRanger (v4.0.0). The reference index was built using the "mkref" function in CellRanger (v3.1.0) and the Ensembl BDGP6.28 *Drosophila melanogaster* genome. The GTF was filtered to remove non-polyA transcripts that overlap with protein-coding gene models, as recommended in the CellRanger tutorial (https://support.10xgenomics.com/single-cell-gene-expression/software/pipelines/latest/using/tutorial_mr). Cell quality filtering and downstream analysis was performed using Seurat (v4.0.5) [176–179] in R (v4.1.1). We began by creating a SeuratObject from the count data using the function "CreateSeuratObject," excluding cells where fewer than 100 genes were detected. Based on the distribution of the genes and transcripts detected per cell, along with the percentage of reads that mapped to mitochondria, we used the following cell-level filters: 24 h dataset, >450 genes/cell, <5,000 genes/cell, >2,500 transcripts/cell, <10% mitochondrial reads/cell; 30 h dataset, >425 genes/cell, <5,000 genes/cell, >1,400 transcripts/cell, <10% mitochondrial reads. This led to the removal of 1,141 (leaving 9,877) and 1,486 (leaving 10,332) cells from the 24 h dataset and 30 h dataset, respectively. A median of 2,083 genes and 11,292 transcripts were detected per cell in the resulting 24 h dataset, but only 1,245 genes and 5,050 transcripts in the 30 h dataset. This discrepancy is due at least in part to the substantially reduced % of reads mapped confidently to the transcriptome in the 30 h sample (24.0% versus 81.1%), the cause of which we were unable to identify but which did not obviously impact the downstream clustering.

## scRNA-seq data clustering

Next, we applied gene-level filtering to the 24 h and 30 h datasets, retaining only those genes expressed in 3 or more cells, and used the function "SCTransform" implemented in Seurat (v4.0.5) to normalize and scale the full 24 h and 30 h datasets [180]. We used 5,000 variable features and regressed out variation due to the percentage of mitochondrial reads. UMAPs of the full datasets were constructed using the "FindNeighbors," "FindClusters," and "RunUMAP" functions using principal components (PCs) 1 to 100 and a clustering resolution of 0.7. To identify doublets produced by stochastic encapsulation of multiple cells within a single bead during the microfluidic process, we used the R package DoubletFinder (v2.0.3) [181]. We observed that when run on the full 24 h dataset, DoubletFinder heavily targeted a subregion of the neuron cluster (approximately 29% of all doublets were in the mechanosensory neuron cluster; S18A Fig). Compared to other cells in the neuron cluster, the identified doublets were heavily enriched for the mitotic marker *string* and cell adhesion gene *klingon*, the latter of which was restricted to the doublet-enriched region of the mechanosensory neuron cluster as well as a subset of GRNs and mechanosensory sheaths (S18B and S18C Fig). Similar targeting of neurons and sheaths by DoubletFinder was observed in the 30h dataset (S18L Fig). Given

this distribution, we reasoned that when run on the entire dataset, DoubletFinder was classifying early differentiating neurons and sheaths, which share a progenitor cell, as doublets. This may be due to their displaying an intermediate transcriptional profile. We therefore ran DoubletFinder separately on the epithelial and nonepithelial datasets. To generate these, we subsetted our data into epithelial cells, identifiable as the large body of contiguous clusters in each dataset (circled in S18A and S18L Fig), and all remaining clusters before reapplying gene-level filtering (retaining those genes expressed in 3 or more cells), rerunning SCTransform (regressing out variation due to the percentage of mitochondrial reads), and reclustering (hereafter all 3 of these processes will be included under the term "reclustering") (Parameters: 24 h and 30 h datasets: epithelial: PCs = 1:150, variable features = 5,000, r = 0.7; nonepithelial: PCs = 1:50, variable features = 5,000, r = 0.7).

At this stage, the nonepithelial cell datasets contained 2,919 (24 h) and 3,163 (30 h) cells. We then further lowered the mitochondrial percent threshold to <5%, which removed a further 15 (24 h) and 38 (30 h) cells, and nonsheath sensory support cells with >2 *nompA* transcripts (24 h: 44 cells; 30 h: 96 cells; S18D, S18E, S18M, and S18N Fig). *nompA* is a known sheath marker [142] and its presence in cells in other sensory support cell clusters indicates possible doublets arising from incomplete dissociation. This left a total of 2,860 (24 h) and 3,029 (30 h) cells, which we reclustered (S18F and S18O Fig) (Parameters: PCs = 1:50, variable features = 5,000, r = 0.7). We then ran DoubletFinder, using the estimated doublet rates outlined in the 10× V3 user guide for the number of cells we recovered (24 h: 8.4%, approximately 11,000 cells; 30 h: 9.3% approximately 11,800 cells; parameters: 1:50 PCs, pN = 0.25, pK = 0.05, without homotypic adjustment). In both datasets, the identified doublets were more evenly dispersed among clusters compared to when run on the full dataset (S18G and S18P). Moreover, genes enriched in the identified doublets were nonspecific and often enriched in epithelial and bract (an identity induced in epithelial cells) clusters (e.g., *CG13023*) (S18H, S18I, and S18Q Fig), suggestive of them being doublets containing epithelial cells. Cells classified as doublets were removed (217 in 24 h; 259 in 30 h). In the 30 h dataset, a further 9 cells that were enriched for the hemocyte marker *NimC4* were removed as presumed doublets from the bract cluster and from the interface between the mechanosensory shaft and socket clusters (S18R Fig). Each dataset was then reclustered (S18J and S18S Fig) (Parameters 24 h: PCs = 1:35, variable features = 5,000, r = 0.7; 30 h: PCs = 1:30, variable features = 5,000, r = 0.7). The 24 h and 30 h nonepithelial datasets were then integrated using the "SelectIntegrationFeatures" (nfeatures = 2,000), "prepSCTIntegration," "RunPCA," "FindIntegrationAnchors" (normalization method = SCT, dims = 1:35, reduction = rpca, k anchor = 5), and "IntegrateData" functions in Seurat. Nonsensory cells (clusters enriched for $NimC4^+$, $repo^+$, $Sox100B^+$, $aos^+$, and $drm^+/vvl^+$) and sensory support cells (clusters enriched for $nompA^+$, $Su(H)^+$, and $sv^+$) were then separately subsetted and reclustered for downstream analysis (nonsensory: PCs = 1:25, r = 0.4; bristle: PCs = 22, r = 0.2). We identified neurons separately in the nonintegrated datasets based on *fne* expression, reclustered them (Parameters 24 h and 30 h: PCs = 1:30, var features = 3,000, r = 1.5; 24 h neuron dataset = 514 cells; 30 h neuron dataset = 473 cells; S18K and S18T Fig), and integrated them separately (same parameters as for the integration detailed above but with k anchor of 20).

For epithelial cells, we ran DoubletFinder and separately reclustered the 2 datasets (Parameters 24 h: PCs = 1:150, variable features = 5,000, r = 0.7, with homotypic adjustment, 6,374 cells; 30 h: PCs = 1:30, variable features = 5,000, r = 0.7, with homotypic adjustment, 6,502 cells). We then integrated the 2 datasets using the approach outlined above (Parameters: nfeatures = 2,000, normalization method = SCT, dims = 1:35, reduction = rpca, k anchor = 5), additionally regressing out variation due to cell cycle stage—specifically, the difference between the G2M and S phase scores (calculated following the steps in the Seurat tutorial;

https://satijalab.org/seurat/archive/v3.0/cell_cycle_vignette.html). The integrated data were then clustered (PCs = 1:30, r = 0.6), the joints removed and clustered separately (PCs = 10, r = 0.2), and the remaining nonjoint cells clustered (PCs = 20, r = 0.4).

### scRNA-seq data analysis

Differential gene expression analyses were conducted on the normalized count data. Data were normalized using the Seurat function "NormalizeData" with normalization method "LogNormalize," where counts for each cell are divided by the total counts for that cell and multiplied by a scale factor (the default of 10,000) and then natural-log transformed using log1p. To perform the differential gene analysis, we used a Wilcoxon rank sum test implemented through the "FindConservedMarkers" function, grouping cells by their dataset of origin (24 h or 30 h) and specifying that genes were only tested if they were present in >10% of the cells in the focal population. Dot plots and plots of gene expression overlaid on UMAPs were generated using the "DotPlot" and "FeaturePlot" functions, respectively.

### Fly Cell Atlas data

To assess the robustness of our developmental neuron classifications in a different, adult dataset, we analyzed the neuronal populations from the FCA leg dataset [38]. This dataset differed in several ways from those we've generated here: Ours is single cell, while the FCA data is single nuclei; ours is pupal (24 h APF and 30 h APF), while the FCA data is from adults; ours is from the first tarsal segment of the foreleg, while the FCA data is from the full length of all 3 pairs of legs. We downloaded the "10× stringent" leg loom data (https://cloud.flycellatlas.org/index.php/s/ZX56j2CcMXnHXYc), from which we extracted the gene expression matrix using the "open_loom" and "get_dgem" functions in the "SCopeLoomR" package (https://github.com/aertslab/SCopeLoomR/). We created a SeuratObject using this data, filtering out genes expressed in <3 cells and cells with <100 genes. The dataset was clustered using the functions described above (PCs = 50, r = 1.7) and neurons identified based on *fne* expression. We then extracted male neurons from this integrated, mixed-sex dataset, which were identifiable from the cell metadata, and reclustered (variable features = 3,000, PCs = 1:25, r = 1).

### Fixation, immunohistochemistry, and microscopy

White P1 prepupae (0 to 1 h APF) were collected and sexed under light microscope based on the presence of testes. Males were then placed on a damp kimwipe and aged in an incubator at 25˚C until the desired time point. Pupae were then placed on their side on sticky tape and a razor blade used to cut away the dorsal half. The cut pupae were then fixed in a 4% paraformaldehyde solution (125 µl 32% PFA, 675 µl $H_2O$, 200 µl 5X TN) for 50 min on a rotator at room temperature and stored at 4˚C. For leg dissections, fixed pupae were removed from the puparia in 1X TNT and tears made in the pupal cuticle at the femur–tibia boundary using forceps. The tibia through to ta5 region was then pulled through the tear, freeing it from the pupal cuticle. At this point, legs expressing fluorescent proteins were mounted in Fluoromount 50 (SouthernBiotech). For immunohistochemistry, the dissected leg region was blocked with 5% goat serum (200 µl 10% goat serum with 200 µl 1X TNT) overnight at 4˚C. Legs were then incubated with primary antibody solution overnight at 4˚C, washed for 21 min 4 times in 1X TNT, incubated with the secondary antibody solution for 2 h at room temperature, washed for 21 min 4 times in 1X TNT, and then mounted in Fluoromount 50 (SouthernBiotech). All stages, from dissection through to staining, were carried out in a glass well. Antibodies were used in solution with 1X TN and a final concentration of 2% goat serum (see Table 1 for identities and concentrations). Confocal images were taken using an Olympus FV1000 laser scanning

confocal microscope or a Zeiss 980 Airyscan. Image stacks were processed using Z-series projection in ImageJ.

## Supporting information

**S1 Fig. Dataset metrics.** (A) A schematic detailing how many cells were filtered out at each stage of processing. Cells were initially filtered from the full dataset and retained based on the number of genes detected per cell (24 h: >450 and <5,000; 30 h: >425 and <5,000), transcripts detected per cell (24 h: >2,500; 30 h: >1,400), and the percentage of transcripts that map to mitochondrial genes (24 h and 30 h: <10%). The datasets were then split into epithelial and nonepithelial cells based on cluster identity (the asterisks here relate to panels (B-E)). Additional filtering was then performed on the nonepithelial cells, removing cells with >5% mitochondrial reads and nonsheath bristle cells in which >2 transcripts of the sheath marker *nompA* were detected, which likely correspond to undissociated doublets. Further doublets were then identified in each dataset using DoubletFinder [181] and removed. In the 30 h dataset, an additional 9 cells positive for the hemocyte marker *NimC4* were identified at the interface between the mechanosensory socket and shaft cluster and at the edge of the bract cluster. These putative hemocyte–bristle cell doublets were also removed. (B-E) Violin plots showing the distribution of (B) genes detected per cell in the 24 h dataset, (C) genes detected per cell in the 30 h dataset, (D) transcripts detected per cell in the 24 h dataset, and (E) transcripts detected per cell in the 30 h dataset. Panels show the distribution in the full datasets after initial filtering based on cell-level quality control metrics (i.e., the distributions at the positions indicated by asterisks in (A). Numerical data with cell barcodes are listed in S3 Data (24 h) and S4 Data (30 h). Code for generating the figure is available at https://www.osf.io/ba8tf. (PDF)

**S2 Fig. Joint markers.** (A) The smallest of the clusters identified in our clustering analysis (Fig 3F) showed highly variable internal expression patterns (i.e., the top markers of this cluster were generally expressed in a relatively small number of its constituent cells). Moreover, this cluster, colored dark green in this plot, showed a markedly higher representation of cells with high mitochondrial read counts, suggesting it may be composed of or enriched for damaged cells. We therefore excluded it from further analysis. Numerical data with cell barcodes are listed in S5 Data. (B-E) UMAP plots of the joint dataset overlaid with the expression of *odd-skipped* family transcription factors. *drm*, *sob*, and *odd* are known to be expressed in the distal edge of each leg segment except tarsal segments 1–4 [52]. Consistent with its widespread expression among epithelial cells in our dataset, *bowl* has been shown to display an overlapping but broader expression pattern (extending into tarsal segments 1–4) than *odd*, *drm*, and *sob* [52]. (F, G) UMAP plots of the full joint and nonjoint epithelial dataset overlaid with the expression of *bowl* (blue) and *drm* (red; F) or *sob* (red; G). (H-V) For each panel, a UMAP plot of the joint dataset (left) and full joint and nonjoint epithelial dataset (right) is overlaid with the expression of a given gene identified during the joint differential gene expression analysis. (H, I) *CG1648* and *Ser* show widespread expression among the proximal tibia/ta1 joint and epithelial cells but are excluded from the other joint clusters. This is expected for *Ser* as $Ser^+$ cells form patterning boundaries in the developing leg that activate joint formation in distally adjacent cells via Notch [193]. (J-N) *Lim1*, *trh*, *caup*, *ara*, and *pdm2* show specific expression in the proximal tibia/ta1 cluster. Of these, *Lim1* is known to be expressed in the tibia where it is required for specification of the tarsus ([194]; see also Fig 5V); $nub^1$ alleles give rise to compromised leg development [195]; *pdm2* shares *cis*-regulatory architecture with *nub*, but, to the best of our knowledge, no role for it in leg development has been characterized [59]; and again, to the best of our knowledge, no roles for the *Iro-C* genes *ara* and *caup* have been

reported in leg development. (O, P) Two of the top DEGs for the proximal tibia/ta1 joint, *CG2016* and *svp*, additionally show enriched expression within a common subregion of what we do not label as joint tissue. Given that this region is *TfAP-2⁻*, it's unlikely that this corresponds to the ta2/ta3 joint and instead more likely corresponds to those cells adjacent to the ta1/ta2 joint. (Q, R) Both *bab* paralogs (*bab1* and *bab2*) are enriched outside of the tibial joint clusters and show particular enrichment within the ta1/ta2 joint cluster, consistent with their known up-regulation in the distal portion of the tarsal segments [56,196]. (S-V) Several of the top DEGs for the ta1/ta2 cluster (*CG12880*, *tnc*, *CG34938*, *GstD1*) do not appear to be specific, rather they show joint-enriched but widespread expression among epithelial cells. Data and code for generating the figure are available at https://www.osf.io/ba8tf.
(PDF)

**S3 Fig. Joint and nonjoint epithelial markers.** (A-K) For each panel, a UMAP plot of the joint dataset (left) and full joint and nonjoint epithelial dataset (right) is overlaid with the expression of a given gene identified during the joint cluster differential gene expression analysis. (A-D) Several of the top differentially expressed genes (DEGs) for the distal tibia/ta1 (*fj*, *Sp1*, *sn*, *Obp56a*) do not appear to be specific, rather they show localized enrichment in joint cells alongside widespread expression in nonjoint epithelial cells. One such gene is *fj*, which we find widely expressed across all joint and nonjoint clusters but enriched in the distal tibia/ta1 cluster (A; Fig 3I). *fj* is known to be required for regional growth along the leg's proximal–distal axis and in imaginal discs shows rings of expression that are complementary to Nub [54,197]. However, our data show a separation between the regions of peak *nub* and *fj* expression (compare Fig 3I with Fig 3D). Several of the remaining top DEGs for the ta1/ta2 (E-G) and distal tibia/ta1 (H-K) joint clusters show more specific expression patterns. (L-R) UMAP plots of the joint dataset overlaid with the expression of genes specifying positional identity. Positional identity is clear at the level of the dorsal–ventral axis (ventral: *wg*, *mid*, *H15;* dorsal: *dpp*, *bi*; L-P), but not anterior–posterior axis (anterior: *ci*; posterior: *hh*; Q, R). (S-AE) UMAP plots of the nonjoint epithelial dataset overlaid with the expression of genes identified as enriched within subregions. Data and code for generating the figure are available at https://www.osf.io/ba8tf.
(PDF)

**S4 Fig. Pupal leg hemocytes form a uniform population.** (A-W) UMAP plots of the nonsensory dataset overlaid with gene expression. (A-N) A selection of hemocyte markers, most of which are part of a cluster of NIM-repeat containing genes on chromosome 2 that also includes the hemocyte-specific *He*. RT-PCR work has previously shown that all of these, except *nimA*, are transcribed in larval hemocytes [69]. We find that all these NIM genes, except *nimA*, are expressed in our hemocyte cluster, as are *Hml* and *srp*, which are known to be expressed in both differentiating and mature plasmatocytes [67,68]. (O-T) Genes identified by Tattikota and colleagues ([70]; see also [72]) as enriched in lamellocytes. We saw no obvious subclustering in relation to these genes: They were either widely expressed among hemocytes (*betaTub60D*, *CG1208*), too patchily expressed among hemocytes to reflect a clear subpopulation (*atilla*, *alphaTub85E*, *CG18754*, *mthl4*), or absent from our dataset entirely (*CG31219*, *CG14610*, *CG15347*, *CG12133*). (U-W) The same study found that crystal cells showed highest enrichment of *PPO1*, *PPO2*, *lz*, *N*, *peb*, and *E(spl)m3-HLH*. As with lamellocytes, we saw no obvious subclustering in relation to these genes: *PPO1*, *PPO2*, and *lz* were absent from our dataset, and *N*, *peb*, and *E(spl)m3-HLH* showed nonspecific expression. We also mapped the top markers of many of the plasmatocyte subclusters identified by Tattikota and colleagues (*Mmp1*, *IM18*, *CecA2*, *CecC*, *Mtk*, *DptB*, *Drs*, *Prx2540-1*, *Prx2540-2*, *CG12896*, *Abl*, *Snoo*, *CG15550*, *CG6023*, *mthl7*, *Cys*, *CG8860*, and *COX8*), but saw no clear subclustering. (X-AG)

Alternating UMAP plots of the full 24 h (X, Z, AB, AD, AF) and 30 h (Y, AA, AC, AE, AG) datasets overlaid with expression of a subset of hemocyte marker genes identified in this study. Data and code for generating the figure are available at https://www.osf.io/ba8tf.
(PDF)

**S5 Fig. Induction of bract cell identity in epithelial cells is accompanied by a remodeling of the transcriptome.** (A-D) Alternating UMAP plots of the full 24 h (A, C) and 30 h (B, D) data-sets overlaid with expression of the bract markers *pnt* and *aos*, EGFR signaling components with known roles in bract formation [74–76]. (E) A dot plot of the top 20 differentially expressed genes from a bract versus nonjoint epithelial comparison. Although the comparison was made between all bracts and all nonjoint epithelial cells, the breakdown per subcluster is depicted ("distal" here refers to the sex comb bearing region). Of these, the ecdysone-induced transcription factor *Eip93F* (also called *E93*), which we found to be up-regulated in bract cells, is known to be expressed in the epithelial cell that will develop as a bract, where it enables *Dll* to respond to EGFR signaling [76]. (F) UMAP plot of the full 24 h dataset overlaid with the expression of the top bract marker, *CG33110*, which encodes a predicted fatty acid elongase. (G) As (F) but in the 30 h dataset, alongside an inset zooming in on the socket cell cluster. The expression of *CG33110* in sockets was more pronounced in the 30 h compared to 24 h dataset. (H) A UMAP plot of the nonsensory dataset overlaid with expression of *CG10348*, a top bract marker. Expression is clearly enriched in bract cells, as well as tendon cells, which are highlighted by a dashed box. (I-P) Alternating UMAP plots of the full 24 h (I, K, M, O) and 30 h (J, L, N, P) datasets overlaid with several top bract markers. In the case of most of these genes, expression is detected outside the bracts and often enriched in the sensory organ cells. Data and code for generating the figure are available at https://www.osf.io/ba8tf.
(PDF)

**S6 Fig. Characterizing the tendon transcriptome.** (A-R) Alternating UMAP plots of the full 24 h (A, C, E, G, I, K, M, O, Q) and 30 h (B, D, F, H, J, L, N, P, R) datasets overlaid with several top tendon markers. (A-N) For *drm*, *Tsp*, *CG13003*, *CG31871*, *yellow-e*, *CG13722*, and *CG9650*, expression was observed in both tendon cells and a subregion of epithelial cells. For *tx* (O, P) and *CG42326* (Q, R), we also observed expression in a subset of shaft and socket cells. In both cases, this socket and shaft expression was more widespread at 30 h. (S) Confocal images of 24 h APF male *trol-GAL4 > UAS-mCD8::GFP* (green) [133] legs counterstained with the neuronal marker anti-Futsch (magenta). The first 3 images show the separate and merged channels from an image of the first tarsal segment. The staining follows much the same pattern showed by anti-Vvl and *1151-GAL4 > UAS-mCherry.nls* in the tendon cells (Fig 4G–4J). Note the concentration of staining around the tibia/ta1 joint, the position of the levator and depressor tendons. The right-hand image shows the distal tibia and proximal ta1 with merged channels. Note the presence of extensive *trol-GAL4* staining in epithelial cells in the tibia—no equivalent epithelial staining was observed in the tarsus. The epithelial *trol-GAL4* staining observed in the tibia was not present in the region proximal to the tibia/ta1 joint. *trol-GAL4* staining was also observed in gustatory receptor neurons (GRNs) (see S12 Fig). (T-V) Expression of *trol* overlaid on the nonsensory (T), full 24 h (U), and full 30 h (V) UMAP plots. Note the expression of *trol* in a subset of GRNs (as shown in (S) and S12 Fig). Some localized expression is present in a region of the epithelial clusters that corresponds to the proximal tibia/ta1 portion of our joint UMAPs, rather than the distal tibia/ta1 region, and therefore likely reflects the tibia/ta1 joint staining rather than that in the more distal tibia, which falls outside of our dissected region. (W-AE) Alternating UMAPs showing expression of known tendon genes *sr*, *Lrt*, and *slow* overlaid on the nonsensory (W, Z, AC), full 24 h (X, AA, AD), and full 30 h (Y, AB, AE) datasets. Note the enriched expression of *sr* in a nontendon region of

the nonsensory UMAP (W). These cells correspond to the *Oaz*+/*repo*+ glia. Data and code for generating the scRNA-seq elements of this figure are available at https://www.osf.io/ba8tf. (PDF)

**S7 Fig. Sox100B+ cells originate within the leg disc in contrast to migratory CNS-derived glia.** (A) A dot plot of the top differentially expressed genes from between-glia comparisons (e.g., *Oaz*+/*repo*+ glia vs. *svp*+/*repo*+ and *Sox100B*+ glia, etc.). (B) L2 larval brain, ventral nerve cord (VNC), and leg disc from a *repo-GAL4* > *UAS-mCherry.nls* (magenta) male counterstained with anti-Sox100B (green). An inset is shown with a close-up of the disc. Note the staining running along the anterior–posterior axis of the VNC and extending up into the brain. In the VNC, this staining falls along the boundary of the neuropil (see S7D Fig). (C) As in (B) but an L3 larval leg disc. (D) The posterior end of the VNC from a 0 h after puparium formation (APF) male stained with both anti-Sox100B (green) and anti-Futsch (magenta), a neuronal marker. Note how anti-Sox100B staining hugs the outer boundary of the neuropil. Data and code for generating the scRNA-seq elements of this figure are available at https://www.osf.io/ba8tf. (PDF)

**S8 Fig. Annotation of glia clusters.** (A, B) 24 h APF male first tarsal segments from *repo-GAL4* > *UAS-mCD8::GFP* (green) counterstained with anti-Futsch (magenta). Note how the staining pattern includes cells hugging the axon trunks, but also, and unlike in the *Lim1-GAL4* staining, cells that branch away from them. This latter feature is particularly clear around the sex comb (see white triangles). Note also that some circular, nonspecific fat body staining is present (examples marked by asterisks in (A)). (C) 24 h APF male forelegs from *1151-GAL4* > *UAS-mCherry.nls* (magenta) males counterstained with anti-Mef2 (green). *1151-GAL4* is a myoblast and tendon cell marker, while Mef2 is a transcription factor essential for cardiac, visceral, and somatic muscle development [203]. Anti-Mef2 staining is restricted to the tibia. (D) Confocal image of a male 48 h pupal leg from a *vkg::GFP* line. Both transmitted light and GFP channels are shown merged. (E) Annotated UMAP plot of nonsensory cells. (F-K) The UMAP shown in (E) overlaid with the expression of: (F) *Jupiter*, a perineural glia marker [96] that here shows widespread expression; (G) *Gs2*, a surface glia marker that here shows patchy expression in the *svp*+/*repo*+ cluster [89]. (H-K) A series of extracellular matrix components that show varying expression profiles across *repo*+ glia, *Sox100B*+ glia, and hemocytes. Data and code for generating the scRNA-seq elements of this figure are available at https://www.osf.io/ba8tf. (PDF)

**S9 Fig. Annotation of sensory neuron populations.** (A-H) In our unsupervised clustering analysis of both the pupal neuron and the FCA neuron datasets, a small subpopulation of mechanosensory neurons clustered separately, labeled in (A) and (E) as *TkR86C*+ mechanosensory neurons. These clusters were enriched for *TkR86C* (A, E), which encodes a receptor for the neuropeptide tachykinin and plays a critical role in male-specific neural, circuits that control aggression [198]. (B-D) In the pupal dataset, these cells were also enriched for the ventral marker *H15*, *CARPA*, and *Side-II*. (F-H) However, in the adult data, only *CARPA* showed any suggestion of being enriched in these cells relative to the other neuron populations. Because of their scarcity, coupled with the absence of strongly specific genes, it's unclear whether the *TkR86C*+ cluster represents a distinct population. We cannot rule out that in the pupal data, they may simply correspond to more developmentally advanced mechanosensory neurons and/or a population from a particular subregion of the leg, the clustering of which is driven by the shared expression of positional markers such as *H15*. (I) A UMAP plot of an

integrated dataset of the 24 h APF male first tarsal segment single-cell RNA-seq data, 30 h APF male first tarsal segment single-cell RNA-seq data, and adult all leg male neuron single-nuclei RNA-seq data. Cells are colored according to the dataset of origin. (J) The UMAP plot given in (I) but this time cells are colored according to the cluster annotation they were assigned based on separate clustering and analysis of the pupal cell and adult nuclei datasets (i.e., those presented in Fig 6A and 6G). Note how the *nvy*$^+$ cluster includes both cells labeled as *nvy*$^+$ gustatory receptor neurons (GRNs) and mechanosensory neurons. Because of this divergent classification, and the broader differences between the datasets in the tissues, their ages, and the dissociation protocol used, we opted to analyze the 2 datasets separately. (K-M) Confocal images of mechanosensory bristles in a subregion of the first tarsal segment from *elav-GAL4 > UAS-mCherry.nls* (magenta) males, counterstained with anti-Pros (green). The channels are shown separately and merged. Note the presence of both Pros$^+$/*elav-GAL4*$^+$ and Pros$^+$/*elav-GAL4*$^-$ cells. This heterogeneity likely reflects between-bristle variation in the developmental stage of mechanosensory sheath cells such that soon after division mechanosensory sheaths are *elav-GAL4*$^+$ (see also [113]). *elav-GAL4* expression is later lost from mechanosensory sheaths, restricting it to neurons. (N) A UMAP plot of the Fly Cell Atlas single-nuclei adult male leg neuron data overlaid with the expression of *CG42566*. Expression of *CG42566* is largely restricted to the 4 gustatory receptor neuron (GRNs) classes. In the pupal data, it is also expressed in a subpopulation of *vvl*$^-$ mechanosensory neurons, which we presume to correspond to the mechanosensory neurons that innervate chemosensory bristles (MSNCBs). It's possible that the patchy expression of *CG42566* in the mechanosensory neuron cluster observed here in the adult data reflects MSNCBs interspersed among mechanosensory neurons. Interestingly, *CG42566* is present in *nvy*$^+$ GRNs in the adult data but absent from them in the pupal data, which may reflect between-GRN developmental timing differences. (O) Confocal images of adult male *eyg-GAL4 > UAS-GFP.S65T* legs. On the left, the first 2 tarsal segments (ta1 and ta2) are shown; on the right, the second through to the fifth (ta2-ta5). Blue arrows and text denote the name and position of visible, stained campaniform sensilla. The naming follows the nomenclature of Dinges and colleagues [8]. Under this nomenclature, the first 3 characters denote the tarsal segment; the fourth, whether the sensilla are found singly (S) or in a group (G) of 2 (as in distal ta1 and ta3) or more (as in ta5); and the fifth, that they are on the front (F) leg. In Ta1GF, note how the cell body of the neuron sits a little further back from the accessory cells, a position it may adopt due to the rotation of the sex comb. Data and code for generating the scRNA-seq elements of this figure are available at https://www.osf.io/ba8tf.
(PDF)

**S10 Fig. Gene expression profiles of gustatory receptor neurons (GRNs) I.** (A-T) UMAPs showing the annotated GRN clusters identified in the integrated pupal neuron data (A) and FCA adult leg data (G) and then overlaid with the expression of genes identified as being specifically expressed or enriched in all GRNs relative to all other neuron populations. (U-AK) The expression of a set of genes that are enriched in the female-sensing and/or male-sensing GRN clusters overlaid on the UMAPs described in (A) and (G). Of these, *ppk23* and *ppk29* are known from previous work to be expressed in both neuron types, while *VGlut* is restricted to female-sensing neurons [21,22,123–126]. We additionally detect the transcription factor *tup* in both populations and *ppk10* and *ppk15* in the male-sensing population. Note that in (AD), 2 of the 3 *ppk15*$^+$ cells outside the main body of M-sensing GRN cells fall within the *acj6*$^+$ region we identified as likely being additional M-sensing cells (see Fig 7AH). Taking the 2 datasets together, *ppk15* represents a strong candidate for a gene involved in male pheromone detection, as does *ppk10*. A gene plotted for just one of the 2 datasets indicates its absence from the

other. Data and code for generating this figure are available at https://www.osf.io/ba8tf.
(PDF)

**S11 Fig. Gene expression profiles of gustatory receptor neurons (GRNs) II.** (A-N) UMAPs showing the annotated GRN clusters identified in the integrated pupal neuron data (A) and FCA adult leg data (G) and then overlaid with the expression of genes identified as being specifically expressed or enriched in the *nvy*+ GRN cluster. (O-T) As well as searching through DEGs, we ran through the expression of all 60 gustatory receptor genes [19,199–202]. Of these, we detected *Gr36a*, *Gr47b*, *Gr58a*, and *Gr63a* exclusively in the pupal ta1 dataset, *Gr64a* exclusively in the adult leg dataset, and *Gr61a* in both. In all cases, expression was limited to just a handful of cells precluding us from confidently assigning *Grs* to specific GRNs. We failed to detect *Gr68a*, a receptor that has been shown to be specifically expressed in the male foreleg [3], in either dataset. (U-X) *Ir52a-d*, the expression of which is shown here overlaid on the UMAP presented in (G), were specifically expressed in the *nvy*+ GRN cluster and only detected in the adult dataset. (Y-Z) The expression of *dsx* appears more widespread among the GRNs in the pupal data compared to the adult data. This may reflect differences in the dissected regions, with *dsx* expression restricted to only the *nvy*+ and *fkh*+ GRNs on the foreleg, while *dsx* is expressed in all *fru*+ neurons regardless of the leg they're on. (AA-AL) A selection of genes enriched specifically in *fkh*+ GRNs or that are shared between *fkh*+ and *nvy*+ GRNs. Note how initially restricted expression in the pupal data of *CAH2* (AA), *jus* (AB), and *Glut4EF* (AF) expands to widespread expression in the adult data (AD-AF). Genes presented for just one of the 2 UMAPs indicates its absence from the other dataset. Data and code for generating this figure are available at https://www.osf.io/ba8tf.
(PDF)

**S12 Fig. Variable expression of the extracellular matrix proteoglycan gene *trol* between gustatory receptor neuron (GRN) classes.** (A) A UMAP showing the annotated GRN clusters identified in the integrated pupal neuron data. (B) The UMAP shown in (A) overlaid with the expression of *trol*, which encodes the extracellular matrix proteoglycan Perlecan. (C) A UMAP showing the annotated GRN clusters identified in the Fly Cell Atlas single-nuclei adult male leg neuron data. (D) The UMAP shown in (C) overlaid with the expression of *trol*. (E-G) Confocal images of 30 h APF male first tarsal segments from *trol-GAL4 > UAS-mCD8::GFP* (green) counterstained with anti-Pros (magenta). We imaged at 30 h APF because at 24 h APF, *trol-GAL4* expression in the bristles of the first tarsal segment was either undetectable or very weak. However, at 24 h APF, *trol-GAL4* expression was clear in the more distal segments, suggesting these bristles may develop ahead of those in the more proximal segments. The imaged region contains 5 chemosensory bristles, which are circled and numbered in (E). anti-Pros marks 5 nuclei (marked with an asterisk) in each bristle: 4 correspond to GRNs and 1 to the sheath. Individual Pros+ nuclei are visible outside of the circled chemosensory bristles shown in (E). These correspond to the sheath cells of mechanosensory (MS) bristles. In (G), note how there is between-bristle variation in the number of *trol-GAL4*+ cells. Bristle 1: 2 strongly positive, 1 weakly positive, 2 negative; Bristle 2: 2 positive, 3 negative; Bristle 3: 2 positive, 3 negative; Bristle 4: 3 positive, 2 negative; Bristle 5: 3 positive, 2 negative. This variability matches the expression profile shown in the UMAPs (B, D), where F-sensing and *fkh*+ GRNs show strong *trol* expression and M-sensing GRNs show variable expression. Data and code for generating the scRNA-seq elements of this figure are available at https://www.osf.io/ba8tf.
(PDF)

**S13 Fig. Potential heterochronic differences between sensory neuron classes.** (A) Annotated UMAP of the pupal integrated neuron data. GRN, gustatory receptor neuron; MSNCB,

mechanosensory neuron in chemosensory bristle. (B) Annotated UMAP of male neuronal cells subsetted from the Fly Cell Atlas single nuclei RNA-seq leg dataset [38]. (C-N) The UMAPs described in (A) and (B) overlaid with a selection of genes identified among the top markers of campaniform sensilla neurons in the pupal data and which show a loss of specificity when moving from the pupal to adult dataset. (O-T) The UMAPs described in (A) and (B) overlaid with a selection of genes identified among the top markers of MSNCBs and/or sex comb neurons in the pupal data and which show a loss of specificity when moving from the pupal to adult dataset. Data and code for generating this figure are available at https://www.osf.io/ba8tf.
(PDF)

**S14 Fig. Gene expression in external mechanotransduction neurons.** (A) Annotated UMAP of the pupal integrated neuron data. GRN, gustatory receptor neuron; MSNCB, mechanosensory neuron in chemosensory bristle. (B) Annotated UMAP of male neuronal cells subsetted from the Fly Cell Atlas single-nuclei RNA-seq leg dataset [38]. (C-AJ) The UMAPs described in (A) and (B) overlaid with a selection of genes showing enriched expression in the different external sensory organ neuron classes involved in mechanotransduction, namely mechanosensory neurons, MSNCBs, sex comb neurons, and campaniform sensilla. (A-M) Genes identified as top markers of mechanosensory neurons in the adult FCA data, but all show expression in other populations. (N-Z) Genes identified as top markers of campaniform sensilla neurons in the adult FCA dataset. Note how many are also expressed in chordotonal organ populations, but few or no mechanosensory neuron populations. (AA-AB) Across both datasets, *Ets65A* appears largely restricted to mechanosensory neurons, MSNCBs, sex comb neurons, and campaniform sensilla. (AC-AD) Although relatively widely expressed in the adult data, *shakB* show marked enrichment in sex comb neurons in the pupal data. (AE-AH) *CG42566* and, to a lesser extent, *CG33639* appear enriched in MSNCBs. (AI-AL) The 2 effectors of sexual differentiation, *fru* and *dsx*, show distinct expression profiles from one another. Data and code for generating this figure are available at https://www.osf.io/ba8tf.
(PDF)

**S15 Fig. Gene expression in putative chordotonal organ neurons.** (A) Annotated UMAP of the pupal integrated neuron data. GRN, gustatory receptor neuron; MSNCB, mechanosensory neuron in chemosensory bristle. (B) Annotated UMAP of male neuronal cells subsetted from the Fly Cell Atlas single-nuclei RNA-seq leg dataset [38]. (C-V) UMAPs of the pupal and adult neurons overlaid with the expression of genes identified as being enriched in the putative chordotonal clusters in the adult data. Note that of these, only *CG3339*, *CG9650*, *CG43689*, and *yuri* were present in the pupal neuron dataset. (C-K) *CG9492*, *CG10185*, *CG13133*, *CG9650*, *nan*, *CG14342*, *Hsc70-2*, and *CG43149* were found across all putative chordotonal organ cells and largely or entirely absent from the pupal ta1 neuron data. (L) *Dhc93AB* was present in all chordotonal clusters except for the *bab1*+ population. (M) *NPFR* was present in all chordotonal clusters except for the *erm*+ population. (N-P) The population enriched for the expression of the transcription factor *erm* showed specific expression of *CG14693*, *Dhc98D*, and *CG13408*. (Q, R) Among chordotonal populations, the transcription factor *yuri* was largely restricted to the *erm*+ cluster but also showed expression in GRNs. (S-V) The *bab1*+ population showed enriched expression of *CG3339* and *CG43689*, although both genes showed spatially restricted expression in a subset of cells in the major chordotonal cluster. Data and code for generating this figure are available at https://www.osf.io/ba8tf.
(PDF)

**S16 Fig. Heterochronic differences between mechanosensory and chemosensory sheath cells.** (A) A subset of the sensory support cell UMAP shown in Fig 8B showing only the mechanosensory and chemosensory sheath populations. Cells are colored in relation to their dataset of origin and therefore, by extension, the time point after puparium formation at which they were collected. (B-J) The UMAP shown in (A) overlaid with the expression of genes in the mechanosensory sheath cluster that we identified as being significantly up-regulated in cells from the 30 h dataset compared to those from the 24 h dataset. Data and code for generating this figure are available at https://www.osf.io/ba8tf.
(PDF)

**S17 Fig. Visualizing the expression of *CG43394*.** Confocal images of a 24 h APF male first tarsal segment from *CG43394-GAL4 > UAS-mCD8::GFP* counterstained with the neuronal marker anti-Futsch. Visible cell bodies are highlighted with arrows in (A). *CG43394* was one of the top markers for a cluster we identified as socket cells from chemosensory bristles. This annotation was based on the strong transcriptomic overlap between the cluster and the other socket populations, including the shared expression of the known socket transcription factors $Su(H)^+$ and $Sox15^+$ (Fig 8G). However, the *CG43394-GAL4 > UAS-mCD8::GFP* staining does not appear to correspond to chemosensory sockets, but rather to a trachea-like channel running through the middle of the tarsal segments. While we cannot exclude the possibility that the cluster we annotated as chemosensory sockets in fact corresponds to these cells, we have reason to doubt that it does. In addition to the socket-like transcriptomic profile of the cells, it seems unlikely that a cell type with approximately 2 cells in the first tarsal segment would generate a cluster of equivalent size to the $CG43394^+$ population. For comparison, this cluster included 69 cells, compared to the 84, 74, and 76 in each of the putative sex comb sockets and shafts and chemosensory sheaths, respectively (approximately 11 of each are found in a single ta1).
(PDF)

**S18 Fig. Processing the 24 h AFP and 30 h APF male first tarsal segment scRNA-seq datasets.** (A) A UMAP plot of the 24 h dataset after low-quality cells have been removed (i.e., those that fail to meet the criteria of >450 genes/cell, <5,000 genes/cell, >2,500 transcripts/cell, <10% mitochondrial reads/cell). Putative doublets identified by DoubletFinder are colored red and singlets blue. Note how in these data, Doublets are heavily enriched within a subregion of mechanosensory neurons and hemocytes. Approximately 29% of cells in the mechanosensory neuron cluster were labeled as doublets. (B) The UMAP plot shown in (A) overlaid with the expression of the mitotic marker *stg*. Note how the same region of the mechanosensory neuron cluster that is enriched for "doublets" is also enriched for *stg*. (C) The UMAP plot shown in (A) overlaid with the expression of *klg*, a gene we identified as one of the top markers of the "doublets" when compared to other mechanosensory cells in the cluster. The only other regions of the UMAP where *klg* expression is detected is in closely associated subsets of mechanosensory sheaths and gustatory receptor neurons (GRNs). By itself, this restricted expression pattern suggests that these cells are not bona fide doublets. Rather, and considered alongside the *stg* expression, it suggests that these cells are early differentiating neurons. Given that neurons and sheaths are formed from the same cell division, it further suggests that *klg* is an early marker of both cell types. (D) The percentage of reads per cell that map to mitochondrial genes overlaid on a UMAP of all the 24 h nonepithelial cells. Note the presence of high % cells at the interface between the socket and shaft clusters. (E) The expression of the sheath marker *nompA* overlaid on a UMAP of all the 24 h nonepithelial cells. Note the presence of $nompA^+$ cells in other sensory support cell clusters (shafts and neurons; see pink arrows). The presence of *nompA* is likely indicative of doublets arising through incomplete dissociation of

these tightly associated cells within sensory bristles. "MS" = mechanosensory. (F) A UMAP plot of all 24 h nonepithelial cells after removal of sensory support cells with >2 *nompA* transcripts detected or where more than 5% of reads mapped to mitochondrial genes. (G) The UMAP shown in (F) with putative doublets identified by DoubletFinder colored red and singlets blue. When run on this subsetted data, the identified doublets are more evenly dispersed among clusters than when run on the full dataset (as shown in A). (H) The UMAP shown in (F) overlaid with the expression of *CG13023*, one of the top markers of the doublets identified in the neuron cluster. *CG13023* is enriched in all areas of the UMAP where doublets were identified as well as in the bract clusters. Bracts develop from epithelial cells, which suggests that the doublets identified are likely to be bona fide doublets that include an epithelial cell. (I) As (H) but overlaid on the full 24 h UMAP shown in (A). *CG13023* is enriched in epithelial cells, further supporting the conclusion that the *CG13023* enriched cells identified by DoubletFinder are bona fide doublets. (J) A UMAP plot of the 24 h nonepithelial cells clustered after the removal of doublets identified in (G). (K) A UMAP plot of the neurons subsetted from (J) and reclustered. (L) A UMAP plot of the 30 h dataset, post-cell filtering, with putative doublets identified by DoubletFinder colored red and singlets blue. As in the 24 h data, doublets are heavily enriched within a subregion of mechanosensory neurons and hemocytes. (M) A UMAP plot of the subsetted nonepithelial cells overlaid with the percentage of reads per cell that map to mitochondrial genes. Note the presence of high % cells at the interface between the socket and shaft clusters. (N) The expression of the sheath marker *nompA* overlaid on a UMAP of all the 30 h nonepithelial cells. As in the 24 h dataset, note the presence of *nompA*$^+$ cells in other sensory support cell clusters (shafts and neurons; see pink arrows). (O) A UMAP plot of all 30 h nonepithelial cells after removal of sensory support cells with >2 *nompA* transcripts detected or where more than 5% of reads mapped to mitochondrial genes. (P) The UMAP shown in (O) with putative doublets identified by DoubletFinder colored red and singlets blue. As with the 24 h data, when run on the nonepithelial subset of cells, the identified doublets are more evenly dispersed among clusters than when run on the full dataset (as shown in (L)). (Q) As in the 24 h data, cells identified as doublets are enriched for *CG13023*, a gene that shows high expression in epithelial cells, from which bracts derive. The full 30 h UMAP shows an equivalent pattern to that observed in the full 24 h UMAP in (I). (R) As shown in (P), DoubletFinder identified several doublets at the interface between the mechanosensory shafts and socket clusters, as well as on the periphery of the bracts. Closer inspection of other cells in this region (see arrows) revealed that many of the neighboring cells identified as singlets expressed high levels of the hemocyte marker NimC4, suggesting that they too were doublets. These cells were manually highlighted and removed. (S) A UMAP plot of the 30 h nonepithelial cells clustered after the removal of doublets identified in (P). (T) A UMAP plot of the neurons subsetted from (S) and reclustered. No campaniform sensilla (i.e., *eyg*$^+$/*toe*$^+$ cells) neurons are readily detected in this UMAP. Data and code for generating this figure are available at https://www.osf.io/ba8tf.
(PDF)

**S1 Data. Cell identities and barcodes for Fig 2L.**
(CSV)

**S2 Data. Cell identities and barcodes for Fig 2N.**
(CSV)

**S3 Data. Transcripts and genes detected per cell barcode in the 24 h dataset after initial cell quality filtering.** Data for S1B and S1D Fig.
(CSV)

**S4 Data. Transcripts and genes detected per cell barcode in the 30 h dataset after initial cell quality filtering.** Data for S1C and S1E Fig.
(CSV)

**S5 Data. Percentage of reads mapping to mitochondrial genes per cell barcode in the epithelial joint dataset.** Data for S2A Fig.
(CSV)

## Acknowledgments

We thank Diana Burkart-Waco, Emily Kumimoto, and Hong Qiu for preparing libraries from our single cell suspensions and performing the sequencing; John Larue, Aidan Angus-Henry, and Tiezheng Fan for assistance with collecting, fixing, and mounting pupae; Kohtaro Tanaka for assistance in developing the first tarsal segment dissection technique; Alex Majane, Evan Witt, and Allison Jevitt for answering questions that helped us design our tissue dissociation protocol; Michael Paddy and Thomas Wilkop for microscopy assistance; and Michel Gho for sharing his observations of Pros expression. Thanks also to Liqun Luo, Cedric Soler, Makoto Sato, David Bilder, Hui-Yu Ku, and the Bloomington Stock Center for *Drosophila* strains, and to the Developmental Studies Hybridoma Bank, Steve Russell, Sarah Certel, Vidya Chandrasekaran, Herbert Jackle, Richard Mann, Marc Freeman, and Megan Corty for antibodies.

## Author Contributions

**Conceptualization:** Ben R. Hopkins, Artyom Kopp.

**Data curation:** Ben R. Hopkins, Artyom Kopp.

**Formal analysis:** Ben R. Hopkins.

**Funding acquisition:** Ben R. Hopkins, Artyom Kopp.

**Investigation:** Ben R. Hopkins, Olga Barmina.

**Methodology:** Ben R. Hopkins, Olga Barmina, Artyom Kopp.

**Project administration:** Artyom Kopp.

**Resources:** Artyom Kopp.

**Software:** Ben R. Hopkins.

**Supervision:** Artyom Kopp.

**Validation:** Ben R. Hopkins, Olga Barmina.

**Visualization:** Ben R. Hopkins.

**Writing – original draft:** Ben R. Hopkins.

**Writing – review & editing:** Ben R. Hopkins, Artyom Kopp.

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
