## [Editor Report · Decision Letter 0]

30 Nov 2022

Dear Dr Hopkins, 

Thank you for submitting your manuscript entitled "Charting the development of Drosophila leg sensory organs at single-cell resolution" for consideration as a Research Article by PLOS Biology.

Your manuscript has now been evaluated by the PLOS Biology editorial staff as well as by an academic editor with relevant expertise and I am writing to let you know that we would like to send your submission out for external peer review.

Once your full submission is complete, your paper will undergo a series of checks in preparation for peer review. After your manuscript has passed the checks it will be sent out for review. To provide the metadata for your submission, please Login to Editorial Manager (https://www.editorialmanager.com/pbiology) within two working days, i.e. by Dec 02 2022 11:59PM.

Kind regards,

Ines

--

Ines Alvarez-Garcia, PhD

Senior Editor

PLOS Biology

---

## [Decision Letter · Decision Letter 1]

22 Feb 2023

Dear Dr Hopkins,

Thank you for your patience while your manuscript entitled "Charting the development of Drosophila leg sensory organs at single-cell resolution" went through peer-review at PLOS Biology. Your manuscript has now been evaluated by the PLOS Biology editors, an Academic Editor with relevant expertise, and by three independent reviewers.

In light of the reviews (attached below), we are pleased to offer you the opportunity to address the comments from the reviewers in a revision that we anticipate should not take you very long - we will leave up to you to perform the experiments suggested by Reviewer 3. We will then assess your revised manuscript and your response to the reviewers' comments with our Academic Editor aiming to avoid further rounds of peer-review.

In addition, we would like to consider your manuscript as a Methods and Resources manuscript, as we think it will fit better this format, thus please select this article type when you submit your revision.

**IMPORTANT - SUBMITTING YOUR REVISION**

3. Resubmission Checklist

a) *PLOS Data Policy*

b) *Published Peer Review*

Sincerely,

Ines

--

Ines Alvarez-Garcia, PhD

Senior Editor

PLOS Biology

Reviewers' comments

Rev. 1:

This manuscript presents a thorough description of cell types that occupy the first tarsal segment of the Drosophila foreleg during pupal development (stages 24h and 30h after pupal formation), in an effort to understand what differentiates the different cell types and how these differences arise during development. Hopkins et al. performed single-cell sequencing of this tissue at the mentioned time points and described the different cells types.

- They first divided the cells into epithelial and non-epithelial cells, as the epithelial cells were very abundant; they discerned the joint cells from the rest of the epithelial cells and identified their transcriptomic signature. They then looked at the rest of the epithelial cells; very interestingly, they saw in the UMAP representation that epithelial cells were organized according to their anatomical position along the anteroposterior, dorsoventral, and proximodistal axes. They found differentially expressed genes that were already known to characterize these axes, as well as new marker genes.

- They then looked into the non-epithelial non-sensory cells, i.e. the hemocytes, glia, and bracts. There, they had a number of interesting observations: first, they noticed that bract cells, despite their epithelial origin, are very different from epithelial cells, concluding that, once committed to a bract identity, the epithelial cells change their transcriptome drastically. Second, and more interestingly, they identified a novel cell type not expressing repo, but resembling glia in many regards (e.g position), that seems to be related with the neural lamella, as it expresses a lamella marker gene (vkg).

- Then, the authors looked into the sensory neurons where they find a very neat combinatorial transcription factor code that discerns all sensory neurons. A lot of work is very well summarized in Figure 9. They use their clustering and annotation to go into detail in the different cell types to study the genes and receptors that differentiate them. One interesting observation that the authors made is that many genes that they find being cluster-specific in pupal cell types are not specific in adult neurons (for probing adult neuron expression, they use the recently published Fly Cell Atlas data). They attribute this to heterochrony, arguing that different neuronal types are at different stages of their differentiation path leading to stage-specific differences that are alleviated when the neurons are fully differentiated.

- Finally, they focused into the support cells (shaft, socket, and sheath) of the different organs, where they found similarities and differences in the expression patterns of homologous support cells of different organs.

In general, the manuscript is very thorough and detailed. It has observations that uncover very interesting biology (such as the Sox100B+ cell type that seems to be associated with the neural lamella) and open new avenues of research. The Introduction is very well written (for non-leg experts, such as myself) and the Discussion is very interesting and stimulating. The Results are very detailed (maybe too detailed for non-aficionados), there is a lot of work and information in a fairly long paper. Importantly, the results are carefully and justly interpreted. The Methods are also very detailed. I think that this paper will be a great addition to the literature and will be very helpful to many future studies. I think it is one of the best written atlas papers I have read.

Having said the above, I believe that the title is exaggerating the findings (or even the experimental setup) of the paper. Profiling these two stages is definitely not sufficient to "chart the development of Drosophila leg sensory organs". I would re-phrase the title to make it more accurate. Similarly, in the abstract, the authors claim to address the question of how this diversity is generated during development, but I don't think that they really address it.

Also, when the authors describe the Sox100B+ cells, they claim that these cells appear to originate with the leg disc itself (line 319). Having seen only static images, it is very hard to conclude this. If the authors want to keep this point, they should try to do live imaging or, at least, a timelapse.

I also have some minor comments, which, I believe, could improve the manuscript, but I acknowledge that there is already enormous amount of work in there, so they are not necessary to be addressed.

- Line 29: "organ" should be "organs"

- Intuitively, GRNs remind me of Gene Regulatory Networks and I was initially confused, although the authors define it immediately. If it is not an already established abbreviation, it might help to change it a bit.

- In line 98, the authors argue reasonably that "techniques and approaches that help overcome these barriers are of significant value". Reading the manuscript and the Methods, it was unclear to me what the authors did differently that enhanced the recovery of the cells that are associated with the cuticle.

- In Figure 2 - Figure supplement 1B,D, it would be good if the y-axes were the same between the two samples for easier comparison.

- In the section where the epithelial cells are annotated based on their anatomical position, the dorsoventral axis overlaps with the proximodistal (both are on the x-axis). It might worth trying a 3D UMAP, in case this resolves the three axes.

- Line 240: "thus, these cells likely correspond to sex comb bracts." An antibody staining could resolve this.

- Line 246-248: This is unclear to me. Does this mean that the support cells are affecting bract cell identity? How? I feel like I am missing something.

- Line 272-273: When is sr usually expressed? An antibody staining or FISH could resolve whether this is indeed due to the developmental stage.

- Line 291-293: Could this be a difference between RNA and protein? Maybe FISH could answer the question.

- Line 332: I am really not a big fan of "data not shown". Could the authors remove this phrase or provide the data or link to a paper?

- Regarding the Sox100B+ cells, the authors argue that these are glia-like cells, although they don't really define what glia is (which is a common problem with glia, as different people define it differently). I was wondering whether the authors looked at other glial markers beyond repo and wrapper.

- The section "repo+ glia and Sox100B+ cells express distinct cell-cell communication gene repertoires" is very handwavy and fits more into the Discussion. Similarly, I feel that the section "Heterochrony in the gene expression profiles of sensory organ neurons" is very preliminary and could also go to the Discussion. Otherwise, the authors would have to justify their claims by performing antibody stainings at different stages to prove that there is indeed heterochrony.

- In Figures 6N-P, it was unclear to me how the authors identified the sheath cells.

- In line 429-430, When is Poxn usually expressed? An antibody staining or FISH could resolve why it was not detected.

- In Figure 7AB, some kind of color-coding of the similarly expressed genes could facilitate the reading of the table.

- Lines 558-560: "But in the adult dataset these differences were reduced or lost, suggesting that the variation we observed between GRNs in the pupal data may reflect heterochronic differences (Figure 7-figure supplement 2AD-AF)." While this is definitely a valid hypothesis, an alternative is that the cells are indeed different during development and they then converge to a similar identity. This has been shown to be the case in other neuronal systems, such as the visual and olfactory neurons.

- Figure 10 seems more appropriate to be in the Supplement.

Rev. 2:

Summary:

In their manuscript "Charting the development of Drosophila leg sensory organs at single-cell resolution" Hopkins et. al improve our understanding of the cell types required to generate the sensory organs in Drosophila tarsi through development. By focusing on the first tarsal segment of the Drosophila pupal foreleg, the authors elucidate the combinatorial transcription factor codes of mechanosensory, sex comb and campaniform sensilla neurons as well as distinguishing between four different gustatory receptor neuron classes. Their work also reveals the receptor and membrane channels found in specific neurons and the transcriptomic differences in the non-neuronal cells in the region, such as tendons, hemocytes, bract cells and glial cells. Additionally they identify a novel cell type that contributes to neural lamella construction. This impressive annotated dataset is also available as a resource for the Drosophila community.

Strengths:

-The authors make excellent use of their developmental timepoint from an academic and experimental perspective. Assaying the first tarsal segment at 24h and 30h APF provides insight into sensory organ specification, while also bypassing the challenge of isolating nuclei protected by the cuticle. By using an enzymatic digestion, the authors are able to improve read depth and reduce gene dropout, benefitting the detection of are cell populations.

-The authors provide a step-by-step understanding of the rationale and process behind their data analysis as they breakdown their datasets by cell type across multiple UMAPS to identify their transcriptional signatures.

-Their work is validated with the use of genetic reagents to confirm the location of the genes and cells being described.

-The comparison to an adult dataset adds to the richness of the resource in tracking sensory organ development in the tarsus.

Questions:

-Based on lines 998-1000, the authors appear to have only included one biological replicate for each of the two timepoints. Is this accurate, and if so, could the authors comment on the rationale behind not including other replicates and the impact they anticipate this may have had on the data? It seems that multiple experimental replicates should be included to provide the basis of a more robust analysis.

-As mentioned in lines 831-839, one possibility for why the authors were only able to identify 4 GRN populations might be because other distinct GRN classes might exist in other tarsal segments, such as in the taste bristles at the tip of the leg. With this in mind, could the authors clarify why they chose to focus on the first tarsal segment to the possible detriment of the other important, but rare, cell types in the rest of the tarsus?

Rev. 3:

This manuscript characterized all major cell types in the male fly foreleg using single-cell RNAseq data from three time points (two from development, generated by this study, and one from adult, published by the FCA project).

This manuscript is probably the most comprehensive paper, I have ever read, for characterizing detailed cell types within a specific tissue. The Authors carefully checked known makers for validation, and also identified new markers, which representing new exciting findings for next studies. For example, they thoroughly compared development and adult data and concluded that there are four types of gustatory receptor neurons. They also identified a novel glia-like cell population which may play important function for the lamella development. All findings are clearly written, and references are properly cited.

I don't have major concerns for this manuscript to publish but have a few suggestions and comments to further improve it.

1. There's no overview of all cell types from the scRNA-seq data, and it's very difficult to follow the analytic logics. I will suggest the authors to re-organize the Fig. 2 into the following orders. First, integrate 24h and 30h datasets and illustrate batch consistency. Second, using one tsne/umap plot to show all broad cell types identified in this study. Marker genes defining each broad cell type should be included in the supplement. Third, show the proportion of the broad cell types from in each dataset, including epithelial joints, epithelial non-joint, neuron, non-sensory, and sensory support cells. Details of different subclustering can be addressed in next figures.

2. Fig. 2A-J, since 24h and 30h data can be integrated well, cell-types markers can be shown in integrated data.

3. Fig 3A and 3C are redundant. Fig 3B and Fig3D are also redundant.

4. For Fig. 3E, the TfAP-2 and nub cluster can be further divided into two sub-cluster. It will be interesting to visualize these two gene expressions together in the same fly to further characterize distal and proximal tibia/ta1 joint cells, (GAL4>nlsRFP, tagged-GFP).

5. Current Fig. 4A and 5A are identical. I suggest the authors highlight cell types that are discussed in Fig. 4 and Fig. 5 accordingly.

6. The authors should provide the general neuronal markers, (elav, nSyb, para) for Fig. 6B.

7. Line 395-396, I don't agree with the statement of "…they integrated poorly". The integration pattern between cells from developmental stages seems very normal to me. From development to adult, there are dramatic transcriptomic changes, even for the same cell types. I consider current pattern in Fig. S6 -1I,J as they integrated well (see Xie et al 2021 eLife; McLaughlin et al 2021 eLife)

8. Fig. 10 should be in the supplementary Figure.

9. There are very limited comparisons between 24h and 30h datasets. It would be great if the authors can compare these two datasets and identify some stage-specific changes. Otherwise, it is not clear why the authors choose these two time points for the experiments.

---

## [Editor Report · Decision Letter 2]

6 Apr 2023

Dear Dr Hopkins,

Thank you for your patience while we considered your revised manuscript entitled "A single-cell atlas of the sexually dimorphic Drosophila foreleg and its sensory organs during development" for publication as a Methods and Resources at PLOS Biology. This revised version of your manuscript has been evaluated by the PLOS Biology editors and by the Academic Editor.

Based on our Academic Editor's assessment of your revision, we are likely to accept this manuscript for publication, provided you satisfactorily address the data requests stated below.

We expect to receive your revised manuscript within two weeks. 

*Published Peer Review History*

*Press*

Sincerely,

Ines

--

Ines Alvarez-Garcia, PhD

Senior Editor

PLOS Biology

Fig. 1L, N; Fig. S1B-E and Fig. S2A

---

## [Editor Report · Decision Letter 3]

3 May 2023

Dear Dr Hopkins,

Thank you for the submission of your revised Methods and Resources entitled "A single-cell atlas of the sexually dimorphic Drosophila foreleg and its sensory organs during development" for publication in PLOS Biology. On behalf of my colleagues and the Academic Editor, Richard Benton, I am delighted to say that we can in principle accept your manuscript for publication, provided you address any remaining formatting and reporting issues. These will be detailed in an email you should receive within 2-3 business days from our colleagues in the journal operations team; no action is required from you until then. Please note that we will not be able to formally accept your manuscript and schedule it for publication until you have completed any requested changes.

PRESS

Sincerely, 

Ines

--

Ines Alvarez-Garcia, PhD

Senior Editor

PLOS Biology
